# Fitness landscapes of human microsatellites

Ryan J. Haasl[1,2]*, Bret A. Payseur[2]*

1 Department of Biology, University of Wisconsin-Platteville, Platteville, Wisconsin, United States of America,
2 Laboratory of Genetics, University of Wisconsin-Madison, Madison, Wisconsin, United States of America

* haaslr@uwplatt.edu (RJH); payseur@wisc.edu (BAP)

## Abstract

Advances in DNA sequencing technology and computation now enable genome-wide scans for natural selection to be conducted on unprecedented scales. By examining patterns of sequence variation among individuals, biologists are identifying genes and variants that affect fitness. Despite this progress, most population genetic methods for characterizing selection assume that variants mutate in a simple manner and at a low rate. Because these assumptions are violated by repetitive sequences, selection remains uncharacterized for an appreciable percentage of the genome. To meet this challenge, we focus on microsatellites, repetitive variants that mutate orders of magnitude faster than single nucleotide variants, can harbor substantial variation, and are known to influence biological function in some cases. We introduce four general models of natural selection that are each characterized by just two parameters, are easily simulated, and are specifically designed for microsatellites. Using a random forests approach to approximate Bayesian computation, we fit these models to carefully chosen microsatellites genotyped in 200 humans from a diverse collection of eight populations. Altogether, we reconstruct detailed fitness landscapes for 43 microsatellites we classify as targets of selection. Microsatellite fitness surfaces are diverse, including a range of selection strengths, contributions from dominance, and variation in the number and size of optimal alleles. Microsatellites that are subject to selection include loci known to cause trinucleotide expansion disorders and modulate gene expression, as well as intergenic loci with no obvious function. The heterogeneity in fitness landscapes we report suggests that genome-scale analyses like those used to assess selection targeting single nucleotide variants run the risk of oversimplifying the evolutionary dynamics of microsatellites. Moreover, our fitness landscapes provide a valuable visualization of the selective dynamics navigated by microsatellites.

## Author summary

Microsatellites are repetitive DNA variants that have a long history of use in genetics. Mounting evidence suggests that some microsatellite variation is adaptive or deleterious. Yet, methods for characterizing natural selection on microsatellites are largely missing, perhaps because microsatellite mutation is complicated. We describe models that capture the relative fitnesses of microsatellite genotypes and enable reconstruction of the microsatellite fitness landscape. Fitting these models to genotypes for 200 individuals in eight

**Data Availability Statement:** Raw and calibrated microsatellite genotypes, simulation code, and R scripts used for computational analyses are available at Dryad Digital Repository: doi:10.5061/dryad.sbcc2frg6.

**Funding:** This research was supported by NIH grants R35GM139412 and R01HG004498 to B.A. P. The funders did not play any role in study design, data collection/analysis, our decision to publish, or in writing the manuscript.

**Competing interests:** The authors have declared that no competing interests exist.

human populations, we identify 43 microsatellites as targets of selection and visualize their fitness surfaces. Our results reveal that selection on microsatellites takes a variety of forms and emphasize the importance of taking mutation into account when considering the fitness of repetitive variation.

## Introduction

Natural selection leaves traces in the genome. By examining patterns of sequence variation among individuals, biologists can find the genes and variants that affect fitness and reveal the form, magnitude, and timing of selection. This realization spurred explosive growth in genome-wide scans for selection [1–5]. Among other discoveries, genes underpinning adaptation have been pinpointed and properties of deleterious mutations have been uncovered. Recent advances in genome sequencing, statistical approaches, and computer simulation promise to deliver broader and deeper insights into the nature of selection.

Despite this notable progress, major gaps persist in our understanding of how selection shapes genomes. One significant deficit is that theory, methods, and data considered in scans for selection are designed for the simplest of variants: single nucleotide polymorphisms (SNPs). For populations that are not too large, the low rate of point mutation justifies the assumption that each SNP observed in a sample arose exactly once somewhere along the sample's genealogy. This assumption, which forms the basis of the influential infinite-sites model [6], allows the development and application of powerful statistical strategies for characterizing selection. Although the focus on SNPs is understandable, it leaves unknown and unexamined the frequency, strength, and mode of selection targeting non-SNP variants.

Genomes are rich in sequences that violate the infinite-sites model. A conspicuous example is the microsatellite, defined as a tandem array of a repeated sequence motif of 1–6 nucleotides (also known as a short tandem repeat). Microsatellites primarily mutate by slipped-strand mispairing of the DNA polymerase [7,8]. This process typically expands or contracts the array by a single repeat, although multi-step expansions and contractions arise with some frequency [9–13]. Because these up-and-down changes in allele size occur at rates that are often many orders of magnitude greater than those of point mutations [9,11,13–15], microsatellite mutation clearly violates assumptions and expectations of mutation under an infinite-sites model: (1) a single microsatellite is subjected to repeated mutational size changes; (2) separate copies of identically-sized alleles at a microsatellite often have independent mutational origins; and (3) a single microsatellite can accumulate considerable allelic diversity.

Other details of microsatellite mutational process and rate widen the departure from the infinite-sites model. Longer microsatellite alleles mutate more rapidly [11–13,16–19]. Both motif size [8,12,19] and sequence [11,15,20] appear to affect microsatellite mutability. The probability that a given mutation will expand or contract current allele size also seems to skew towards contraction as allele size increases [11]. Despite these complexities, the considerable frequency of microsatellites in genomes across the tree of life suggests these common variants warrant careful consideration for potential adaptive and maladaptive contributions to organismal fitness. According to estimates in humans, microsatellites constitute between 3% [21] and 5% [22] of the genome, contribute 76–85 *de novo* mutations to each child [23], and impose a burden of deleterious mutations on par with that of SNPs [24].

Although microsatellite polymorphisms were originally used as neutral markers useful for reconstructing population history and inferring relatedness [25], there is ample evidence that some microsatellite variation has functional consequences [26,27]. Runaway repeat expansions

at long microsatellites located in exons or regulatory sequences cause or increase risk for 63 human diseases, including cancer and a variety of neurodegenerative and neurodevelopmental disorders [28,29]. Probands with autism spectrum disorder carry more *de novo* mutations at microsatellites than their unaffected siblings [12]. Genotypes at coding microsatellites observed to harbor *de novo* mutations are associated with a range of human quantitative traits, including morphology, cognition, cardiovascular function, metabolism, behavior, and respiration [30]. Microsatellite polymorphisms shape morphology of the skull and limbs in dogs [31], social behavior in voles[32], circadian rhythms in flies [33], genetic incompatibilities among lines of *Arabidopsis thaliana* [34], and cell adhesion and biofilm formation in budding yeast [35].

One of the ways microsatellites are likely to change phenotypes is by altering gene expression. Eukaryotic cis-regulatory elements are enriched for microsatellites, with as many as 25% of budding yeast promoters harboring them [36]. Microsatellites show evidence of activating or repressing transcription in budding yeast [36], flies [37], mice [38], and humans [39–42]. Genotypes at thousands of microsatellites are associated with gene expression differences in humans [41,43,44], and a subset of these expression-QTL (eQTL) are connected to complex traits such as schizophrenia and inflammatory bowel disease [44]. Although the molecular mechanisms by which microsatellites affect gene expression are still being elucidated, recent experiments demonstrate that microsatellites located in cis-regulatory sequences can change binding affinities of transcription factors by more than 70-fold [22]. Microsatellites also have potential to regulate gene expression by modulating nucleosome positioning [45,46], impacting DNA methylation [47], and forming non-canonical secondary structures in DNA or RNA [48,49]. Microsatellite mutation causes minute changes to the length and content of a DNA sequence; data increasingly suggest these changes can be of real functional consequence when they occur in microsatellites located in gene control regions.

The functional capacity of microsatellites suggests a role for natural selection at these loci. With mutation moving allele sizes up or down in small increments, a single microsatellite could generate an array of quantitative trait values on which selection could act to confer adaptation [31,50–57]. More generally, variation at microsatellites with functional consequences can be pictured as a balance between recurrent mutation and selection against deleterious alleles [12,58].

Development of methods to infer the contributions of natural selection to microsatellite evolution faces multiple barriers. First, in contrast to inferences about selection on SNPs, conclusions about selection on microsatellites cannot be drawn without accounting for the effects of a high rate of recurrent mutation and other mutational realities described above. Moreover, theoretical results specific to microsatellites that might guide this accounting are lacking, with some notable exceptions [59–62]. Second, microsatellite fitness surfaces are more complicated than SNP fitness surfaces. A locus with $n$ alleles can generate $g_n = (n)(n+1)/2$ diploid genotypes. Therefore, the relative fitness of a diallelic SNP is summarized by just three values. This situation contrasts sharply with the reasonable scenario of a microsatellite harboring eight alleles and, therefore, up to 36 distinct genotypes that might each have a different relative fitness as members of a rugged fitness landscape. Finally, genotyping structural variants such as microsatellites is more difficult than genotyping SNPs. Although the accuracy of microsatellite genotyping from short-read sequencing continues to improve [27,63–65], and long-read sequencing offers potential for further gains [66], the genotyping error rate is non-trivial. In particular, "heterozygote dropout" in which heterozygotes at a microsatellite are called as homozygotes continues to hamper accurate genome-scale acquisition of microsatellite genotype data [12]. Due to these challenges, the characteristics of natural selection targeting microsatellites remain mostly unknown several decades after these loci were first introduced into population genetics.

Haasl and Payseur [58] developed models of microsatellite evolution that enable rapid forward-simulation of population samples under selective and neutral conditions while incorporating the mutational complexity inherent to microsatellites. Using an approximate Bayesian computation (ABC) approach, we reconstructed the fitness surface of a GAA microsatellite that causes Friedreich's ataxia in humans, estimating that genotypes containing alleles beyond the length associated with disease suffer severe fitness reductions of ~90% [58]. These methodological and inferential advances ushered in a few studies with the explicit goal of characterizing selection on microsatellites. Recently, others adopted the simulation approach of [58] to build methods for quantifying the strength of selection on single microsatellites [12] and inferring the distribution of selection coefficients for groups of microsatellites [24]. Applying their method assuming a single model of selection to 62,941 microsatellites genotyped from short-read sequencing in European humans, Mitra et al. [12] detected selection on many loci, with most estimated selection coefficients ranging between 0.001 and 0.01 (though simulations suggested limited power to detect weak selection). Inferences based on 86,327 microsatellites genotyped from short-read sequencing in Europeans suggested substantial variation in selection coefficients among loci, with the overall selective burden of *de novo* microsatellite mutations estimated to exceed that for standing microsatellite polymorphism [24].

Basic questions about the nature of selection on microsatellites remain unanswered following these initial efforts to characterize it. How strong is selection targeting microsatellites with complex fitness landscapes? Does selection on microsatellites target a single optimum allele size or are there multiple optima, as stark bi-modal and even multi-modal distributions of allele frequencies at some loci suggest? What is the role of dominance among a set of microsatellite alleles with different marginal fitnesses?

Focusing on microsatellites carefully chosen for their potential to affect fitness, we address these and related questions. We describe a variety of selective and neutral models tailored to microsatellite variation. To assess the fit of these models to genotype data from a diverse collection of eight human populations, we use approximate Bayesian computation with random forests (ABC-RF)[67]. For the first time, we characterize the detailed fitness landscapes of microsatellites inferred to be evolving under one of four selection models. We argue that the fitness landscape metaphor is well-suited to intuiting and evaluating the role of selection at loci with complex mutational dynamics.

## Materials and methods

### Data collection

DNA samples of 200 individuals participating in the 1000 Genomes Project [68] were acquired from Coriell Cell Laboratories (Camden, NJ). Individuals belonged to eight 1000 Genomes Project populations: 25 CEPH/Utah (CEU), 27 Han Chinese in Beijing, China (CHB), 30 Finnish in Finland (FIN), 25 Gujarati Indian in Houston, TX (GIH), 25 Luhya in Webuye, Kenya (LWK), 18 Mexican ancestry in Los Angeles (MXL), 25 Toscani in Italy (TSI), and 25 Yoruba in Ibadan, Nigeria (YRI). Of these populations, two are native African (LWK, YRI), two are native European (FIN, TSI), and one is native Asian (CHB). The remaining three populations consisted of dispersed individuals whose ancestry is European (CEU), Asian (GIH), and an admixture of European and Native American (MXL).

Genotypes of 146 microsatellite loci were determined for all 200 individuals at Prevention Genetics (Marshfield, WI) using fluorescent fragment-length analysis. Sanger sequencing was used to corroborate the genotypes of individuals at ten randomly chosen loci. In all cases, sequences agreed with the genotype calls derived from fragment-length analysis. None of the

sequenced microsatellites at these loci harbored point or indel imperfections to the microsatellite sequences. Two of the 146 loci harbored point imperfections in the human genome reference sequence.

Genotyped loci comprised mostly dinucleotide and trinucleotide repeats, although several mononucleotide, tetranucleotide, and pentanucleotide loci were included. Loci were further classified as: (1) potentially targeted by natural selection, (2) intergenic, or (3) closely linked to known targets of selection on SNPs. We chose members of the first category based on information in the literature and genomic databases, including potential for mutation to affect protein sequence as well as functional and suggestive evidence of regulatory activity [39, 69, 70]. Microsatellites that were not chosen based on findings in the literature were instead chosen based on the inherent hazard their mutation presents (e.g., coding dinucleotides) or their proximity to indicators of gene regulatory activity such as DNAseI hypersensitivity marks and putative cis-regulatory elements. We also included four trinucleotide loci previously identified as having anomalously low variance for their allele size in the human genome (Jim Weber, personal communication). Finally, we included loci implicated in trinucleotide repeat disorders. Although sampled variation at these loci was largely non-pathogenic, we were interested in testing whether non-pathogenic distributions are shaped by selection. Intergenic loci were restricted to microsatellites >250kbp away from the closest RefSeq gene based on RefSeq Release 47 released immediately prior to genotyping. Although our results suggest otherwise in several cases (see Results), we chose these loci assuming they had a greater chance of evolving neutrally and therefore providing points of comparison to loci we suspected might be targeted by selection. In this manuscript, we report results for loci suspected to be targeted by selection as well as putatively neutral, intergenic loci. Results for loci linked to known targets of selection will be reported elsewhere.

Conversion of fragment-length genotypes (in base pairs) to repeat-length genotypes was made possible by comparing fragment-length genotypes to previously predicted genotype calls (in repeat-length) for 1000 Genomes individuals using LOBSTR [71,72]. Although LOBSTR-based calls of the very same loci as we genotyped allowed the desired conversion, it also revealed that publicly available microsatellite genotype data [72] are enriched for false identification of heterozygotes as homozygotes. This issue is likely due to the difficulties associated with calling microsatellites based on short-read DNA sequence data, particularly the tendency of next-generation sequencing technologies to miss evidence of the second allele in a heterozygote [73] (S1 Text provides an illustrative example).

## Evolutionary modeling

Simulation models of molecular evolution capture necessary details of the evolutionary process by including parameters related to mutation, migration, neutral/non-neutral evolution, etc. When simulations of the model are run, parameter values are drawn from prior distributions. We used uniform priors whose minimum and maximum represent the bounds of reasonable values a parameter can take (see subsection *Natural Selection* below).

Approximate Bayesian computation (ABC) methods require a large pool of simulated results because the unifying core of these methods is a comparison of the one empirical dataset to many simulated datasets. Following simulation, both empirical and simulated data sets are summarized using summary statistics–e.g., allele frequencies, observed and expected heterozygosity, number of private alleles, etc. Posterior probability densities on model parameters are approximated using the parameters from those simulations whose summary statistics best match those of the empirical data.

## Mutation

Direct, empirical estimates of human mutation rate as a function of allele size were obtained from [11] for dinucleotide microsatellites. Dinucleotide mutation rate was $\mu_{di} = (0.08+0.002a +0.0002a^2)/1000$, where $a$ is allele size. To our knowledge, the literature does not provide similarly informative estimates of mutation rates specific to *allele size* for mono-, tri-, tetra-, and penta-nucleotide microsatellites in human. To address uncertainty in microsatellite mutation rates, we used several mutation curves in simulations of each motif size to integrate over this uncertainty (S1 Fig). However, we caution that loci that depart considerably from mean rates of microsatellite mutation for a given motif may provide results that are less robust than those presented here. Mutation rates at microsatellites with allele sizes <8 tend to be quite low and, to our knowledge, have not been estimated for microsatellites of any motif size. Therefore, we assumed mutation rates of alleles of size <8 to be between $10^{-8}$ and $10^{-6}$ (S1 Table).

## Demography

Because demographic models specifying the split times of 1000 Genomes populations are not available, we devised a general demographic model based on recent literature. The timing and order of population splits is shown in S2 Fig. We assumed a generation time of 20 years and that all populations had an equilibrium effective population size of $N_e$ = 10,000. The initial population sizes of descendant populations in our model are expressed as proportions of the parent population. For example, descendant populations B and C might start at 0.99 and 0.01 the size of ancestral population A; this represents a peripatric split. In other cases, both descendant populations were established as small samples of their ancestral population. For example, both CEU and TSI begin at 0.01 the size of their ancestral population, WECA (S2 Fig). These low proportions reflect the fact that CEU and TSI are not truly sister populations descended from a single population split. All descendant populations grew linearly, adding 100 individuals per generation until they reached the equilibrium $N_e$ of 10,000.

## Natural selection

We devised five models of microsatellite evolution derived from our earlier work on the topic [58]. In addition to a neutral model, four models of natural selection were used. Under the neutral model, all genotypes were assigned a relative fitness of 1.0. The models of natural selection hinge on key allele size, $\alpha$, which is the allele with the greatest marginal fitness (= 1.0). Two of the four models of selection are *single-optimum* models because the genotypic fitness surface includes a single genotype ($\alpha/\alpha$) with maximum relative fitness of 1.0. The other two models of selection are *periodic-optima* models, in which all alleles with a size that is a multiple of $\alpha$ have a marginal fitness of 1.0. Thus, genotypes $\alpha/\alpha$, $\alpha/2\alpha$, $2\alpha/2\alpha$, $2\alpha/3\alpha$, and so forth all have a relative fitness of 1.0 under the periodic optima models.

We set the bounds of the uniform prior for $s$ to (0, 0.02]. The lower limit on the prior for $s$ excluded zero to ensure some selective pressure. The upper limit of 0.02 was based on preliminary simulations and early analyses. Although we initially simulated greater values of $s$, we set the upper limit on $s$ to 0.02 when running our final analyses. This decision was based on the prior knowledge that values of $s > 0.02$ were never recovered in 95% highest posterior densities, making simulations with values of $s$ this high unnecessary for consideration with our data set. Bounds on the uniform prior distribution of key allele size $\alpha$ were set to [2, 50]. Here, the upper bound was set to 50 because the greatest mean allele size at any of the loci studied was ~36 repeats long. We therefore reasoned that any locus in our data set targeted by selection would not have a key allele size greater than 50.

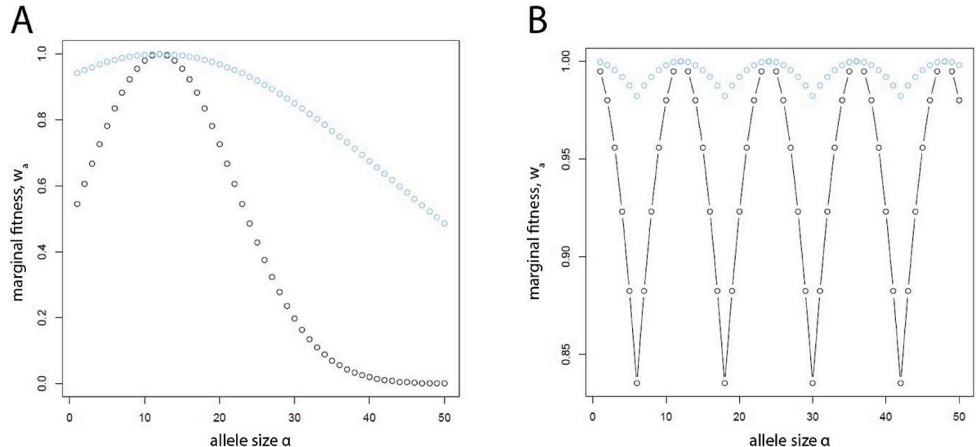

**Fig 1. Marginal fitness vs. allele size.** (A) Single-optimum, with key allele size of $\alpha$ = 12 and $s$ = 0.005 (black circles) or $s$ = 0.0005 (blue circles). (B) Periodic-optima using the same parameter values, with marginal fitness of 1.0 for multiples of the key allele size– 12, 24, 36, 48, etc.

The periodic-optima models were motivated by the frequent occurrence of bimodal allele frequency distributions and the occasional occurrence of trimodal frequency distributions among the loci we sampled and genotyped. We note that fitness landscapes inferred under the periodic-optima models imply fitness peaks at genotypes far from the empirical distribution of genotypes in our sample. We advise that this subset of fitness peaks should be treated with caution.

The marginal fitness of an allele of size $a$ under the two single-optimum models was calculated using a Gaussian fitness function [74]: $w_a = e^{-s(a-\alpha)^2}$, where $s$ is the strength of selection. Thus, two parameters– $\alpha$ and $s$–defined marginal fitness (Fig 1A). Under the two periodic-optima models, we defined the marginal fitness of an allele of size $a$ as:

$w_a = exp\left(-s\left[min\left(a - \alpha\lfloor\frac{a}{\alpha}\rfloor, \alpha\lceil\frac{a}{\alpha}\rceil - a\right)\right)\right]$, where $\lfloor\rfloor$ and $\lceil\rceil$ are the floor and ceiling operators, respectively (Fig 1B). Again, the same two parameters defined the models of marginal fitness.

In addition to the marginal fitness of each component allele, the relative fitness of a genotype depended on whether the model was additive or dominant. Under the *additive* single-optimum (ASO) and periodic-optima (APO) models, the relative fitness of a genotype was set to the average marginal fitness of the two component alleles. Thus, under the ASO model, genotype $\alpha/\alpha$ would have a relative fitness of 1.0 while *any* other genotype would have a lower relative fitness.

Under the *dominant* single-optimum (DSO) and periodic-optima (DPO) models, the relative fitness of a genotype was set to the maximum marginal fitness of the two component alleles–i.e., genotypes $\alpha/\alpha$, $\alpha/(\alpha-2)$, $\alpha/(\alpha+3)$, and any other genotype with at least one allele of size $\alpha$ would be assigned a relative fitness of 1.0. Fig 2 shows representative genotypic fitness surfaces for the four selection models.

## Simulation

**Simulation mechanics.** We expanded the forward-in-time simulation program FOR-TUNA [75] to implement microsatellite mutation. To prepare for prospective simulation using FORTUNA, we produced a pool of starting allele frequency distributions (AFDs) that we could draw from randomly to provide different starting points for the universal common ancestor (UCA) population (S2 Fig). To produce this pool of starting AFDs, we ran coalescent

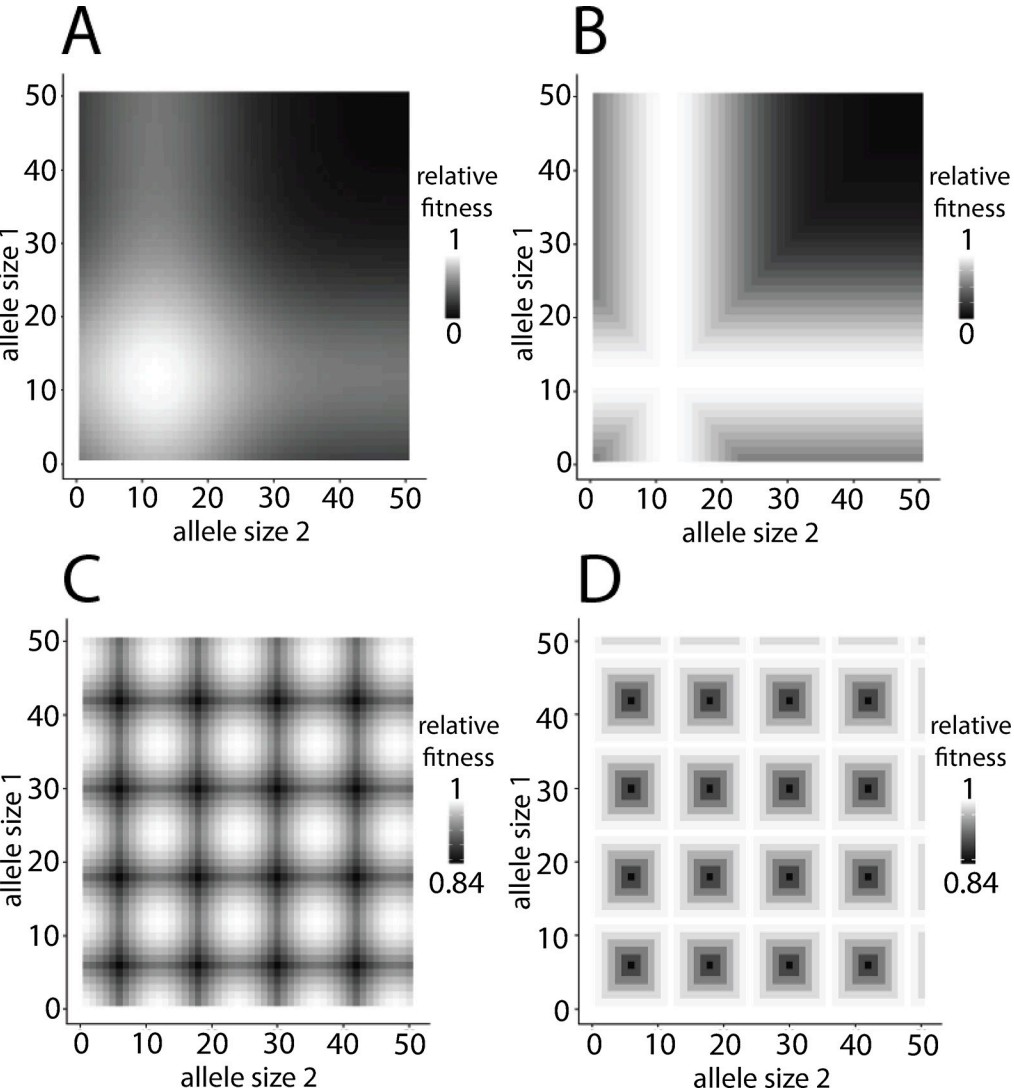

**Fig 2. Genotypic fitness surfaces for the four models of natural selection simulated.** Parameter values used to generate these graphs were $\alpha = 12$ and $s = 0.005$. (A) additive, single-optimum (ASO) with a single, most-fit genotype of 12/12. (B) dominant, single-optimum (DSO) where all genotypes 12/_ have a relative fitness of 1.0. (C) additive, periodic-optima (APO). (D) dominant, periodic-optima (DPO). Compare the relative fitness scales of the ASO (A) and DSO (B) models to those of APO (C) and DPO (D); the relative fitness of the least-fit genotype on the surface is substantially lower under the single-optimum models.

simulations implemented using the R package *coala* [76] with MS [77] as the backend simulation program.

To ensure a diversity of AFDs, we drew random variates from two random variables, which served as arguments to the coalescent simulation: (1) the population mutation rate $\theta = 4N_e\mu$ from a uniform real distribution on the range [0.04, 40], which–assuming $N_e = 10,000$ –correspond to individual mutation rates on the range $[10^{-3}, 10^{-6}]$, and; (2) the ancestral allele size of the simulated coalescent genealogy. Respectively, these two variables influenced the dispersion and mean of resulting AFDs. We generated 20,000 AFDs; among these simulated distributions, variance in allele size spanned [0, 240] with an average $V_{AS}$ equal to 2.24, while mean allele size spanned [4.56, 37.47] with average $M_{AS}$ equal to 15.72.

The demographic model in S2 Fig was encoded within a FORTUNA parameters file (see Data Availability), and each forward-in-time simulation proceeded as follows:

1. Draw a random starting AFD from the pre-simulated pool of AFDs.

2. Use allele frequencies of the chosen AFD as probabilities, and randomly generate 10,000 diploid genotypes assuming Hardy-Weinberg equilibrium. These genotypes represent the UCA population at generation 0 of the simulation.

3. Draw one of the five models of microsatellite selection, each with a probability of 0.2.

4. If the neutral model is drawn, set the relative fitness of all genotypes to 1.0. Else, draw random values of $\alpha$ and $s$ from their prior distributions and calculate the marginal fitness and relative fitness of each allele and genotype, respectively.

5. Simulate (forward-in-time) the UCA population for 2,000 generations; the UCA evolves neutrally even if natural selection will be applied to descendant populations.

6. Split the UCA population. Randomly assign members of the UCA population to its descendant YRI and non-West-African Common Ancestor (NWACA) populations at the proportions shown in S2 Fig. No gene flow occurs between the descendant populations; they now evolve separately from each other.

7. Carry out all subsequent demographic events shown in S2 Fig as the simulation proceeds forward in time, generation by generation.

8. 4,750 generations after the first split, halt the simulation.

9. Generate a simulated data set that has the same structure as the empirical data set described above–i.e., randomly sample a number of individuals from each of the eight populations equal to the number sampled in the empirical data set (e.g., 18 from MXL).

10. Summarize the data set as the final output of the simulation. Specifically, calculate 408 summary statistics (*features*, in the context of random forests) from the sampled genotypes. For each of the eight populations calculate the frequencies of alleles size 1–50 plus observed heterozygosity among sampled individuals. (8x51 = 408 summary statistics).

In a single generation, the following steps were taken *for each population* in existence that generation:

1. Identify the population size of the next generation ($N_{e_{t+1}}$). If $N_{e_t} = 10,000$, then population size remains constant and $N_{e_{t+1}}$ also equals 10,000. Otherwise, effective population size is linearly increasing towards 10,000. FORTUNA pre-calculates the population size schedule of each population, thereby providing the value of $N_{e_{t+1}}$ when $N_{e_t} \neq 10,000$.

2. (Natural selection models only) Draw a random, real-valued variate $x$ from ~Uniform (0,1) for each member of the population. If $x \leq w_{a_1/a_2}$, where $w_{a_1/a_2}$ is the relative fitness of an individual's genotype, the individual remains in the population as a potential parent. Otherwise, the individual is removed from the population.

3. Choose pairs of parents of the next generation by sampling with replacement from the (surviving) members of the population.

4. Sample one of the two alleles of each parent with equal probability.

5. Check for germline mutation in the first inherited allele by drawing a random, real-valued variate $m$ from ~Uniform (0,1). If $m \leq \mu$ –where $\mu$ is the appropriate mutation rate for the

size of the considered allele and the type of microsatellite (e.g., dinucleotide) being simulated–change the size of the inherited allele. Choose the size of change (i.e., step size) based on observations of [11]. With a probability of 0.5, multiply the change in allele size by -1. Add the change to current allele size to obtain the allele size of the inherited allele.

6. Repeat step 5 for the second inherited allele.

7. Add the genotype/individual to the next generation's population.

8. Repeat steps 3–7 until the number of individuals in the next generation equals the population size scheduled for this population in the current generation according to the demographic model.

9. Check if this population is set to split this generation. If so, generate the two descendant populations by randomly sampling the appropriate proportion of individuals from those simulated for the next generation. Eliminate the ancestral population from continued simulation. Proceed to the next generation.

**Simulation in practice.**   Following the simulation procedure detailed in the previous subsection, we generated a *reference table* where each entry/observation/row comprised (1) summarized simulation results (the 408 features) paired with (2) the randomly drawn parameter values $\alpha$ and $s$ used in the simulation. Reference tables included the results of 500,000 simulations each for the two motif sizes present in the empirical data–namely, dinucleotide and trinucleotide repeats. Because forward-in-time simulators require considerable time and memory relative to coalescent simulators, we distributed the computational load of simulations across a cluster available to us through the University of Wisconsin-Madison Center for High Throughput Computing.

## ABC-random forests analysis

Like all approximate Bayesian computation (ABC) methods, the ABC using random forests (ABC-RF) approach provides likelihood-free approximations to posterior distributions of competing models or parameter values by comparing simulated and empirical data. ABC-RF differs from traditional ABC methods by using the machine learning RF algorithm to learn the connection between summary statistics and a model or parameter. This approach contrasts with the simple ABC rejection algorithm [78, 79], which requires calculation of the distance $\delta$ between each simulated data set and the empirical data set. The level of approximation is then controlled by setting a tolerance level $\epsilon$, and only simulations where $\delta \leq \epsilon$ are retained for approximation of the posterior density. We used both the regression and classification variants of random forests in ABC-RF for parameter estimation and model choice, respectively.

Because ABC-RF uses all training examples (i.e., the collection of simulated data referred to as a *reference table*), the method makes efficient use of simulation effort and eliminates the need for selecting only those simulations that match based on a semi-arbitrary tolerance. Moreover, the out-of-bag error calculated as a standard procedure in random forests analysis may be used as a good estimate of test error [80]. This eliminates the need to produce extra simulations in the service of cross-validation. Nevertheless, ABC-RF remains an ABC method because the observations used to train the RF model are *summaries of simulated data*, which substitute for explicit likelihood calculation.

**Model choice.**   Robert et al. [81] showed that posterior probabilities on competing models estimated using ABC are sensitive to whether the summary statistics are statistically *sufficient*– i.e., the summarized data set provides as much information about the data as the full data set.

Unfortunately, the sufficiency of summary statistics can be difficult to assess, and it is reasonable to assume that we usually fail to capture all information of inferential merit in a complex data set when we summarize it. The authors also showed that this issue risks calculation of Bayes factors that are highly divergent from their true values.

Pudlo et al. [67]later developed an ABC approach to model choice that uses the ensemble machine learning approach of random forests (RFs). RFs are used in both classification and regression contexts, depending on whether the outcome variable is discrete (such as models to choose between) or continuous (such as real-valued parameter values to estimate). ABC-RF model choice sidesteps the statistical issues raised by Robert et al. [81] by restructuring the task of model choice as a classification problem addressed using RFs [67]. Specifically, RFs learn the relationship between summary statistics and models or parameters and avoid the statistically dubious calculation of Bayes factors when using ABC approaches that calculate distance metrics between simulated and empirical data instead. ABC-RF does not require calculation of Bayes factors and we used a non-informative prior on the model chosen for simulation, further reducing influence of prior choice on model choice.

**Parameter estimation.** Subsequently, Raynal et al. [80] extended the application of random forests to ABC-based estimation of parameter values. In this case, random forests are constructed in a regression context, as the outcome (parameter) values are numeric.

**ABC-RF in practice.** A separate RF model was trained for each microsatellite motif size. We tuned three hyperparameters of known importance to RF performance: (1) number of trees in each forest (*ntree*), (2) the number of randomly selected features to use at each split in a tree (*nfeat*), and (3) the minimum number of simulated observations required to perform another split (*minobs*) when constructing a decision tree component of the random forests algorithm. Hyperparameter tuning for a particular motif size was performed using five-fold cross-validation and a reference table of results from simulations of the relevant motif size. In this way, we identified the combination of hyperparameters that provided the smallest test error for a particular motif size. For all motif types, tuned hyperparameters were found to be *ntree* = 200 and *minobs* = 15. Depending on motif type, the tuned value of *nfeat* was either 20 or 30.

To train an RF model for a particular motif size, we passed the tuned hyperparameter values to an RF classification algorithm and trained the RF model using a reference table of 500,000 observations derived from simulations in which the relevant motif size was assumed (Fig 3A). Empirical data from each locus of the same motif size were input to the trained RF model, which returned the posterior probabilities of each evolutionary model. If a locus received a plurality or majority of support for one of the four models of natural selection, we then approximated the posterior distributions of the two selection parameters: key allele size $\alpha$ and strength of selection $s$.

We assessed RF model performance using out-of-bag (OOB) error estimation. Random forests are a collection of decision trees, each trained on a bootstrap replicate of the original data. We followed the common practice of setting aside roughly one-third of the bootstrap replicates (the OOB data) to assess performance. The other two-thirds of bootstrap replicates were used to train the model. Model performance was assessed by measuring how well the trained model predicted the model and parameters used to simulate the data set in question.

We used the R package abcrf [80] to estimate parameters $\alpha$ and $s$. We trained a regression RF model for each combination of motif size and evolutionary model; these regression RF models were trained on a reference table of 100,000 results from simulations in which the relevant combination of motif size and evolutionary model were assumed (Fig 3B). Before training each model, hyperparameter tuning was performed as above for model choice. Each empirical locus was input to the appropriate trained RF model, returning the approximated posterior distribution of the parameter of interest. Per [80], each parameter was estimated separately (Fig 3B).

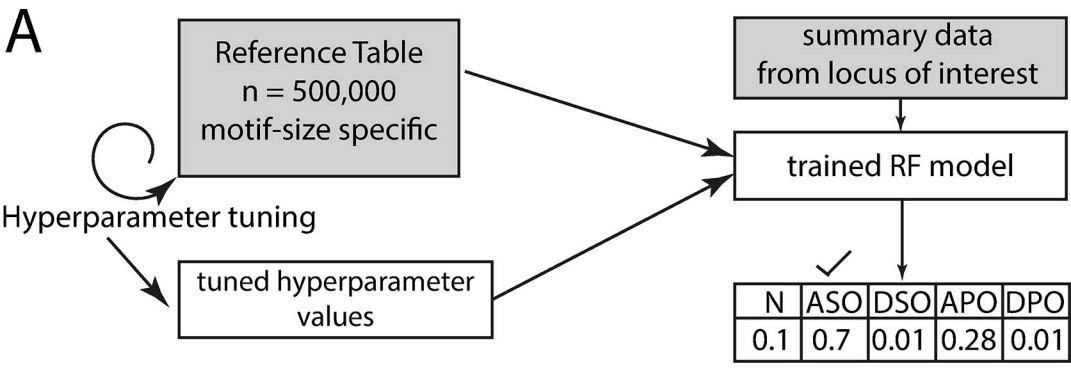

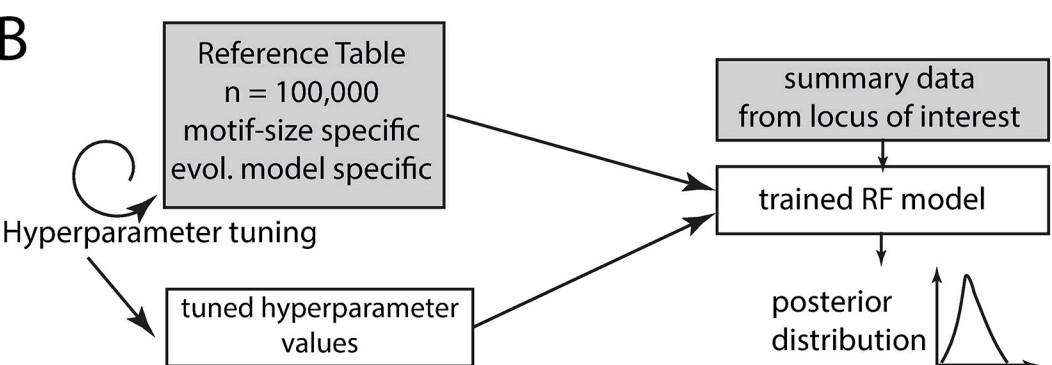

**Fig 3.** (A) Model choice using the ABC-RF classification algorithm. A separate RF model was trained for each motif size based on 500,000 simulations. Output for a single locus is an array of posterior probabilities for each evolutionary model. We chose the model with the greatest posterior probability. (B) Parameter estimation using the ABC-RF regression algorithm. A separate RF model was trained for each motif size and each model of natural selection based on 100,000 simulations. The output for a single locus is the approximated posterior density of the parameters.

Raw and calibrated microsatellite genotypes, simulation code, and R scripts used for computational analyses are available in a Dryad Digital Repository [82].

## Results

### Microsatellites are multifaceted genetic variants

To visualize patterns of variation in our sample, we began by comparing mean allele size ($M_{AS}$) with variance in allele size ($V_{AS}$) for a class of microsatellites suspected *a priori* to evolve neutrally: 54 intergenic dinucleotide loci. Most of these microsatellites were CA repeats, while 10 were TA repeats. We found that $V_{AS}$ *generally* increased with allele size (Fig 4A), most likely due to the increased mutability of longer microsatellites [11–13]. Indeed, we have previously used this well-known characteristic of microsatellites to show that a simple summary of variation such as the number of alleles at a locus assumed to evolve neutrally does well at estimating the population mutation rate $\theta$ [83].

Yet, even in this modest sample of intergenic dinucleotides we found conspicuous outliers, where $V_{AS}$ was much greater than the general increasing trend. In most cases, these outliers bore inflated $V_{AS}$ due to bimodal allele size distributions. This includes the locus with the greatest $V_{AS}$ in this dinucleotide set, located on chromosome 4, with a striking bimodal allele frequency distribution, and one which our method classified as evolving under natural

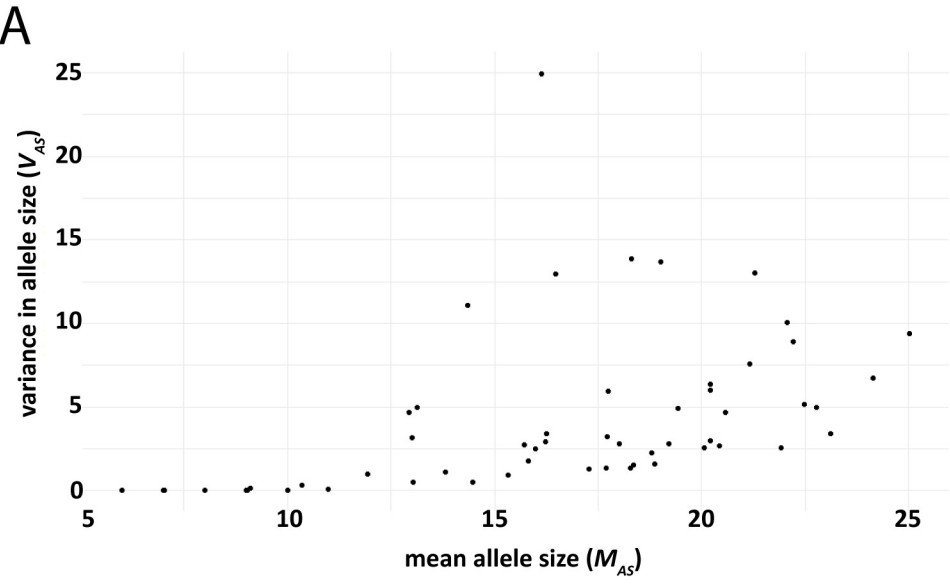

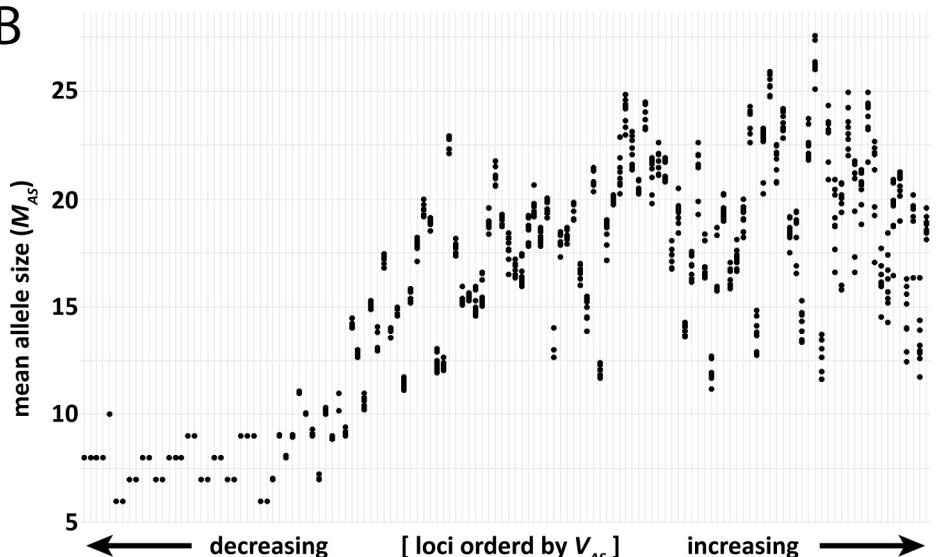

**Fig 4.** (A) Sample-wide variance in allele size ($V_{AS}$) vs. mean allele size ($M_{AS}$) for 54 intergenic dinucleotide microsatellites. In general, genetic variance at microsatellites increases with allele size, despite clear outliers with anomalously high variance. (B) Mean allele size of the same 54 intergenic dinucleotide microsatellites. Loci, from left to right along the x-axis, are ordered by increasing variance in allele size. Each locus is represented by eight dots, which are the means of each sampled population for that locus.

selection (see final case study below). Using a different visualization of the data, Fig 4B shows that while there was a general trend of increasing variation with allele size, many loci defied the trend. Many of the intergenic dinucleotide microsatellites with relatively large $V_{AS}$ had surprisingly small mean allele sizes. Furthermore, some loci showed substantial variation in mean allele size among populations, while others showed little or none.

Therefore, our carefully chosen sampling of microsatellite variation demonstrates the unpredictability of the microsatellite diversity landscape, and how little we know about the

numerous factors shaping these multifaceted allele frequency distributions. Even when we grouped microsatellites by motif size, the microsatellites in these groupings were difficult to justify as identically distributed. This suggests more formal stochastic models of natural selection of microsatellite alleles would likely require a hierarchical modeling approach.

## Method performance

**Random forests models reliably distinguish microsatellites under selection from neutral microsatellites.** We developed an approach based on approximate Bayesian computation random forests (ABC-RF) to probabilistically assign microsatellites to one of four selection models (additive, single-optimum; additive, periodic-optima; dominant, single-optimum; or dominant, periodic-optima) or a neutral model. We first tested the performance of our method using simulated datasets. The method did remarkably well at distinguishing between loci simulated under neutral models and models of selection. Loci simulated under the neutral model were misclassified as one of the four selection models only 6.2% (dinucleotides) or 5.1% (trinucleotides) of the time. Conversely, loci simulated under one of the four selection models were misclassified as neutral loci only 3.1% (dinucleotides) or 3.2% (trinucleotides), or 5.1% (tetranucleotides) of the time (S2 Table). The ability of the method to accurately choose a particular selection model among all five models reached 66% and varied depending on the model and motif type (S2 Table). Importantly, most classification errors were misidentifications where data simulated under an additive model were classified as evolving under a dominant model or vice versa (S2 Table). These results demonstrate that our method reliably distinguishes between selection and neutrality and chooses among selective models with an accuracy that is reasonable given the complexity of microsatellite evolution.

**The magnitude of polymorphism is the primary determinant of the evolutionary model chosen.** We assessed the importance of each summary statistic of microsatellite variation to model choice using the metric of Gini decrease (see Methods). Of the 408 summary statistics used to train RF models for each motif size, the eight summary statistics of greatest variable importance were the values of observed heterozygosity in each sampled population. A large gap between the importance of observed heterozygosity and allele frequencies for model choice suggests the *overall level of genetic variation* (as summarized by observed heterozygosity) holds greater power to distinguish models of microsatellite evolution than the more granular allele frequency data.

## Identifying and characterizing selection on microsatellites

**Microsatellites in different genomic and functional contexts show evidence of natural selection.** We applied our method to a carefully chosen set of 94 microsatellites genotyped in 200 humans from a diverse collection of eight populations. Microsatellites were chosen based on established or predicted effects on disease, gene expression, or protein sequences. We also included microsatellites with previous indications of aberrant patterns of variation and intergenic microsatellites not suspected to be functional.

Our results for different categories of microsatellites again emphasize the multifaceted nature of microsatellites and point to causes of distinct patterns of variation among loci. Genomic context provides certain expectations for how a microsatellite should evolve and thereby influences the need to invoke natural selection as an explanatory factor of the variation observed at a given locus. For example, exonic dinucleotides are clear hazards, as one-step and two-step mutations to these microsatellites are also frameshift mutations. Here, the genomic context would seem to necessitate a role for purifying natural selection, which is supported by the near-total lack of variation at these loci in our sample (Table 1) [84]. As a counterexample,

**Table 1. Microsatellites analyzed and posterior probabilities assigned to each evolutionary model.** Locus categories include causative of a known trinucleotide repeat disorder (triDisorder), other trinucleotide with low variance and relatively large mean allele size (triOther), Rockman and Wray (2002) site (RW), DNase I hypersensitivity site (DNaseI), intergenic dinucleotide (InterDi), and coding dinucleotide (CodingDi). Observed variance in allele size ($V_{AS}$) and mean allele size ($M_{AS}$) are global–i.e., calculated using genotypes pooled from all eight populations sampled. N = neutral; ASO = additive, single-optimum; DSO = dominant, single-optimum; APO = additive, periodic-optima; DPO = dominant, periodic-optima. The evolutionary model with the greatest posterior support is bold-faced for each locus, and loci receiving greatest support from a model of selection are backshaded. Loci are ordered by motif size, then by category, and finally by $V_{AS}$.

| position (GRCh38/hg38) | gene | category | $M_{AS}$ | $V_{AS}$ | motif size | EVOLUTIONARY MODEL SUPPORT | | | | |
|---|---|---|---|---|---|---|---|---|---|---|
| | | | | | | N | ASO | DSO | APO | DPO |
| chr6:170561908–170562021 | *TBP* | triDisorder | 36.46 | 3.04 | trinucleotide | 0.225 | 0.095 | 0.15 | 0.235 | **0.295** |
| chr12:6936729–6936773 | *ATN1*† | triDisorder | 17.6 | 8.122 | trinucleotide | **0.82** | 0.015 | 0.1 | 0.015 | 0.05 |
| chr19:45770205–45770264 | *DMPK* | triDisorder | 10.76 | 19.867 | trinucleotide | **0.98** | 0.005 | 0.005 | 0 | 0.01 |
| chr14:92071011–92071034 | *ATXN3* | triDisorder | 16.54 | 31.968 | trinucleotide | **0.79** | 0.025 | 0.04 | 0.025 | 0.12 |
| chr12:13774980–13775006 | *GRIN2B* | triOther | 9.06 | 0.061 | trinucleotide | 0.015 | **0.465** | 0.04 | 0.435 | 0.045 |
| chr1:168798149–168798174 | *DPT* | triOther | 9.98 | 0.108 | trinucleotide | 0.045 | 0.305 | 0.08 | **0.37** | 0.2 |
| chr9:137597929–137597955 | *ZYMND19 / ARRDC1* | triOther | 9.08 | 0.109 | trinucleotide | 0.05 | **0.385** | 0.105 | 0.305 | 0.155 |
| chr19:49154627–49154656 | *HRC* | triOther | 11.32 | 0.298 | trinucleotide | 0.005 | 0.275 | 0.205 | **0.35** | 0.165 |
| chr2:218381902–218381945 | *SLC11A1*‡ | RW | 22.26 | 0.202 | dinucleotide | 0 | **0.315** | **0.315** | 0.15 | 0.22 |
| chr2:233760234–233760248 | *UGT1A1* | RW | 7.39 | 0.26 | dinucleotide | **0.55** | 0.115 | 0.165 | 0.08 | 0.09 |
| chr7:134461143–134461192 | *AKR1B1* | RW | 23.97 | 1.636 | dinucleotide | **0.495** | 0.02 | 0.195 | 0.095 | 0.195 |
| chr16:67433688–67433728 | *HSD11B2* | RW | 19.17 | 2.101 | dinucleotide | **0.345** | 0.07 | 0.255 | 0.05 | 0.28 |
| chr7:55020560–55020593 | *EGFR* | RW | 18.8 | 3.96 | dinucleotide | **0.935** | 0.01 | 0.025 | 0.005 | 0.025 |
| chr7:94396316–94396346 | *COL1A2* | RW | 22.12 | 4.056 | dinucleotide | 0.3 | 0.06 | 0.15 | 0.11 | **0.38** |
| chr16:10183569–10183608 | *GRIN2A* | RW | 24.2 | 10.006 | dinucleotide | **0.955** | 0 | 0.005 | 0 | 0.04 |
| chr4:153702763–153702807 | *TLR2* | RW | 20.46 | 10.524 | dinucleotide | **0.955** | 0 | 0.005 | 0.005 | 0.035 |
| chr20:46008773–46008818 | *MMP9* | RW | 19.26 | 11.324 | dinucleotide | **0.715** | 0.075 | 0.075 | 0.08 | 0.055 |
| chr12:117361840–117361899 | *NOS1* | RW | 26.02 | 16.313 | dinucleotide | **0.76** | 0.01 | 0.08 | 0.03 | 0.12 |
| chr2:222299318–222299368 | *PAX3* | RW | 21.29 | 28.02 | trinucleotide | **0.49** | 0.035 | 0.12 | 0.12 | 0.235 |
| chr22:37150087–37150138 | *ILR2B* | DNaseI | 26.4 | 5.814 | dinucleotide | **0.97** | 0.005 | 0.01 | 0 | 0.015 |
| chr4:46389803–46389862 | *GABRA2* | DNaseI | 29.78 | 18.95 | dinucleotide | **0.945** | 0 | 0.03 | 0 | 0.025 |
| chr2:122694258–122694269 | *intergenic* | InterDi | 6 | 0 | dinucleotide | 0 | **1** | 0 | 0 | 0 |
| chr4:179569040–179569052 | *intergenic* | InterDi | 6 | 0 | dinucleotide | 0 | **1** | 0 | 0 | 0 |
| chr4:179870055–179870068 | *intergenic* | InterDi | 7 | 0 | dinucleotide | 0 | **1** | 0 | 0 | 0 |
| chr11:23562933–23562948 | *intergenic* | InterDi | 8 | 0 | dinucleotide | 0 | **0.995** | 0 | 0.005 | 0 |
| chr18:68152140–68152155 | *intergenic* | InterDi | 8.01 | 0.005 | dinucleotide | 0 | **0.71** | 0.22 | 0.07 | 0 |
| chr2:123157113–123157132 | *intergenic* | InterDi | 10.01 | 0.012 | dinucleotide | 0 | 0.375 | 0.01 | **0.6** | 0.015 |
| chr13:103886021–103886035 | *intergenic* | InterDi | 7.03 | 0.028 | dinucleotide | 0.175 | 0.265 | **0.345** | 0.07 | 0.145 |
| chr6:23404523–23404540 | *intergenic* | InterDi | 9.09 | 0.165 | dinucleotide | 0.01 | 0.215 | 0.205 | 0.295 | **0.275** |
| chr2:220893214–220893233 | *intergenic* | InterDi | 10.34 | 0.274 | dinucleotide | 0.03 | 0.23 | 0.185 | **0.31** | 0.245 |
| chr13:34477912–34477943 | *intergenic* | InterDi | 13.04 | 0.425 | dinucleotide | 0.18 | 0.18 | 0.205 | 0.2 | **0.235** |
| chr8:59646798–59646824 | *intergenic* | InterDi | 13.2 | 0.449 | dinucleotide | 0.105 | 0.245 | 0.16 | **0.305** | 0.185 |
| chr2:22727950–22727977 | *intergenic* | InterDi | 14.46 | 0.508 | dinucleotide | 0.015 | 0.05 | 0.11 | **0.525** | 0.3 |
| chr4:36922294–36922325 | *intergenic* | InterDi | 15.35 | 0.93 | dinucleotide | 0.2 | 0.09 | 0.23 | 0.145 | **0.335** |
| chr14:25650631–25650654 | *intergenic* | InterDi | 11.95 | 3.98 | dinucleotide | **0.365** | 0.06 | 0.1 | 0.23 | 0.245 |
| chr5:165163389–165163415 | *intergenic* | InterDi | 13.82 | 1.001 | dinucleotide | 0.02 | **0.35** | 0.12 | 0.345 | 0.165 |
| chr9:11326956–11326990 | *intergenic* | InterDi | 18.29 | 1.27 | dinucleotide | 0.33 | 0.14 | **0.415** | 0.045 | 0.07 |
| chr5:4291710–4291743 | *intergenic* | InterDi | 17.7 | 1.297 | dinucleotide | 0.155 | 0.215 | **0.46** | 0.045 | 0.125 |
| chr2:5169497–5169531 | *intergenic* | InterDi | 17.26 | 1.326 | dinucleotide | **0.37** | 0.145 | 0.26 | 0.07 | 0.155 |
| chr18:78132609–78132645 | *intergenic* | InterDi | 18.34 | 1.49 | dinucleotide | **0.835** | 0.02 | 0.095 | 0 | 0.05 |
| chr4:180292583–180292610 | *intergenic* | InterDi | 15.82 | 1.548 | dinucleotide | **0.99** | 0 | 0 | 0.005 | 0.005 |

*(Continued)*

**Table 1.** (Continued)

| position (GRCh38/hg38) | gene | category | $M_{AS}$ | $V_{AS}$ | motif size | EVOLUTIONARY MODEL SUPPORT | | | | |
|---|---|---|---|---|---|---|---|---|---|---|
| | | | | | | N | ASO | DSO | APO | DPO |
| chr14:86420038–86420078 | *intergenic* | InterDi | 18.87 | 1.588 | dinucleotide | 0.16 | 0.18 | **0.515** | 0.04 | 0.105 |
| chr18:77757670–77757700 | *intergenic* | InterDi | 15.95 | 1.948 | dinucleotide | **0.335** | 0.09 | 0.185 | 0.13 | 0.26 |
| chr3:78148236–78148273 | *intergenic* | InterDi | 18.77 | 2.233 | dinucleotide | **0.995** | 0 | 0 | 0.005 | 0 |
| chr12:115237887–115237931 | *intergenic* | InterDi | 21.92 | 2.325 | dinucleotide | **0.725** | 0.04 | 0.11 | 0.04 | 0.085 |
| chr4:111085123–111085165 | *intergenic* | InterDi | 20.43 | 2.37 | dinucleotide | **0.6** | 0.07 | 0.195 | 0.03 | 0.105 |
| chr18:67988182–67988220 | *intergenic* | InterDi | 20.07 | 2.568 | dinucleotide | **0.705** | 0.04 | 0.19 | 0.02 | 0.045 |
| chr18:77756408–77756447 | *intergenic* | InterDi | 19.21 | 2.673 | dinucleotide | **0.655** | 0.035 | 0.095 | 0.035 | 0.18 |
| chr20:12563271–12563308 | *intergenic* | InterDi | 18.05 | 2.698 | dinucleotide | **0.705** | 0.035 | 0.21 | 0 | 0.05 |
| chr4:189361924–189361956 | *intergenic* | InterDi | 15.73 | 2.713 | dinucleotide | **0.985** | 0 | 0.01 | 0 | 0.005 |
| chr21:19737378–19737418 | *intergenic* | InterDi | 20.22 | 2.802 | dinucleotide | **0.415** | 0.06 | 0.185 | 0.075 | 0.265 |
| chr4:111679317–111679363 | *intergenic* | InterDi | 16.22 | 2.846 | dinucleotide | **0.725** | 0.015 | 0.11 | 0.04 | 0.11 |
| chr2:122878526–122878562 | *intergenic* | InterDi | 17.75 | 3.058 | dinucleotide | **0.775** | 0.03 | 0.15 | 0.005 | 0.04 |
| chr9:107899569–107899612 | *intergenic* | InterDi | 23.1 | 3.071 | dinucleotide | **0.975** | 0 | 0.01 | 0.005 | 0.01 |
| chr2:220276568–220276591 | *intergenic* | InterDi | 13.03 | 3.195 | dinucleotide | 0.065 | 0.045 | 0.19 | 0.295 | 0.405 |
| chr3:176147805–176147835 | *intergenic* | InterDi | 16.29 | 3.417 | dinucleotide | **0.305** | 0.085 | 0.245 | 0.1 | 0.265 |
| chr2:122478653–122478692 | *intergenic* | InterDi | 20.18 | 4.34 | dinucleotide | **0.92** | 0.02 | 0.025 | 0 | 0.035 |
| chr5:166163816–166163861 | *intergenic* | InterDi | 19.44 | 4.559 | dinucleotide | **0.855** | 0.01 | 0.03 | 0.015 | 0.09 |
| chr4:178999316–178999360 | *intergenic* | InterDi | 20.6 | 4.583 | dinucleotide | **0.985** | 0 | 0 | 0 | 0.015 |
| chr4:179791304–179791340 | *intergenic* | InterDi | 12.96 | 4.692 | dinucleotide | **1** | 0 | 0 | 0 | 0 |
| chr21:19689224–19689246 | *intergenic* | InterDi | 13.13 | 4.901 | dinucleotide | **0.995** | 0 | 0 | 0 | 0.005 |
| chr2:22801784–22801832 | *intergenic* | InterDi | 20.22 | 5.252 | dinucleotide | **0.875** | 0.015 | 0.04 | 0.015 | 0.055 |
| chr1:208700324–208700368 | *intergenic* | InterDi | 21.2 | 5.299 | dinucleotide | **0.86** | 0.02 | 0.055 | 0.01 | 0.055 |
| chr2:220601670–220601702 | *intergenic* | InterDi | 17.72 | 5.842 | dinucleotide | **0.465** | 0.03 | 0.125 | 0.09 | 0.29 |
| chr1:105150363–105150421 | *intergenic* | InterDi | 24.14 | 6.336 | dinucleotide | **0.965** | 0.005 | 0.01 | 0 | 0.02 |
| chr9:12251507–12251554 | *intergenic* | InterDi | 22.23 | 7.672 | dinucleotide | **0.995** | 0 | 0 | 0.005 | 0 |
| chr6:14491991–14492043 | *intergenic* | InterDi | 25.05 | 8.952 | dinucleotide | **0.93** | 0.01 | 0.02 | 0 | 0.04 |
| chr10:2444326–2444377 | *intergenic* | InterDi | 22.07 | 9.763 | dinucleotide | **0.98** | 0 | 0.005 | 0 | 0.015 |
| chr4:179223249–179223284 | *intergenic* | InterDi | 14.37 | 10.304 | dinucleotide | **0.58** | 0.07 | 0.115 | 0.08 | 0.155 |
| chr12:115615779–115615830 | *intergenic* | InterDi | 21.32 | 11.893 | dinucleotide | **0.975** | 0 | 0.005 | 0.005 | 0.015 |
| chr2:34160602–34160643 | *intergenic* | InterDi | 20.79 | 11.939 | dinucleotide | **0.97** | 0 | 0.01 | 0 | 0.02 |
| chr1:104442028–104442061 | *intergenic* | InterDi | 16.52 | 12.321 | dinucleotide | **0.855** | 0.01 | 0.03 | 0.02 | 0.085 |
| chr10:9032228–9032267 | *intergenic* | InterDi | 18.33 | 13.56 | dinucleotide | **0.735** | 0.025 | 0.05 | 0.05 | 0.14 |
| chr4:179230364–179230386 | *intergenic* | InterDi | 16.13 | 24.661 | dinucleotide | 0.125 | 0.05 | 0.205 | 0.195 | **0.425** |
| chr1:181710966–181710979 | *CACNA1E* | CodingDi | 8 | 0 | dinucleotide | 0 | **0.995** | 0 | 0.005 | 0 |
| chr1:241683527–241683540 | *WDR64* | CodingDi | 7 | 0 | dinucleotide | 0 | **1** | 0 | 0 | 0 |
| chr1:42536529–42536545 | *CCDC30* | CodingDi | 8 | 0 | dinucleotide | 0 | **0.995** | 0 | 0.005 | 0 |
| chr2:10777693–10777706 | *ATP6V1C2* | CodingDi | 7 | 0 | dinucleotide | 0 | **1** | 0 | 0 | 0 |
| chr3:65387175–65387188 | *MAGI1* | CodingDi | 7 | 0 | dinucleotide | 0 | **1** | 0 | 0 | 0 |
| chr4:53453081–53453094 | *FIP1L1* | CodingDi | 7 | 0 | dinucleotide | 0 | **1** | 0 | 0 | 0 |
| chr5:140653008–140653021 | *IK* | CodingDi | 7 | 0 | dinucleotide | 0 | **1** | 0 | 0 | 0 |
| chr5:6754900–6754913 | *PAPD7* | CodingDi | 7 | 0 | dinucleotide | 0 | **1** | 0 | 0 | 0 |
| chr7:139409619–139409632 | *LUC7L2* | CodingDi | 7 | 0 | dinucleotide | 0 | **1** | 0 | 0 | 0 |
| chr10:128106848–128106862 | *MKI67* | CodingDi | 7 | 0 | dinucleotide | 0 | **1** | 0 | 0 | 0 |
| chr14:23059293–23059306 | *ACIN1* | CodingDi | 7 | 0 | dinucleotide | 0 | **1** | 0 | 0 | 0 |

*(Continued)*

**Table 1.** (Continued)

| position (GRCh38/hg38) | gene | category | $M_{AS}$ | $V_{AS}$ | motif size | N | ASO | DSO | APO | DPO |
|---|---|---|---|---|---|---|---|---|---|---|
| | | | | | | **EVOLUTIONARY MODEL SUPPORT** | | | | |
| chr16:50334767–50334780 | BRD7 | CodingDi | 7 | 0 | dinucleotide | 0 | **1** | 0 | 0 | 0 |
| chr22:35265582–35265595 | HMGXB4 | CodingDi | 7 | 0 | dinucleotide | 0 | **1** | 0 | 0 | 0 |
| chr4:1025267–1025304 | FGFRL1 | CodingDi | 10 | 0 | dinucleotide | 0.015 | 0.175 | 0.015 | **0.715** | 0.08 |
| chr3:128573476–128573493 | C3ORF27 | CodingDi | 9.01 | 0.002 | dinucleotide | 0 | **0.62** | 0.07 | 0.305 | 0.005 |
| chr15:58890365–58890378 | SLTM | CodingDi | 6.99 | 0.007 | dinucleotide | 0.015 | 0.17 | **0.525** | 0.125 | 0.165 |
| chr10:68749528–68749541 | CCAR1 | CodingDi | 6.99 | 0.02 | dinucleotide | 0.04 | **0.44** | 0.39 | 0.055 | 0.075 |
| chr3:50118455–50118476 | RBM5 | CodingDi | 10.95 | 0.093 | dinucleotide | 0 | 0.355 | 0.03 | **0.495** | 0.12 |
| chr10:49326926–49326961 | C10ORF71 | CodingDi | 22.81 | 4.715 | dinucleotide | **0.99** | 0 | 0.005 | 0.005 | 0 |

† The CAG repeat in *ATIN1* contains two point imperfections: CAG CA<u>A</u> CAG CA<u>A</u> (CAG)[15] in the reference human genome. The 19 trinucleotide repeats code for a 19-redisdue-long poly-Q tract.

‡ The CA repeat in *SLC11A1* contains four point imperfections: (CA)[9] <u>CG</u> <u>TA</u> (CA)[4] <u>CG</u> <u>TA</u> (CA)[5].

the genomic context of an ATG-trinucleotide microsatellite in *GRIN2B* is an intron of the gene, suggesting evolution of the locus should be largely unrestricted by selective pressures. Yet, as shown below, this microsatellite harbored nearly no variation. Here, the *lack* of variation at a site suggests the influence of natural selection because genomic context leads to the expectation of appreciable variation.

To determine which microsatellites are targets of selection, empirical summary statistics for each locus were input to RF models trained on simulations of the appropriate motif size. 43 of the 94 loci (45.7%) tested received greatest posterior support for a selection model over the neutral model (Table 1).

Three of the four trinucleotide microsatellites implicated in triplet expansion disorders—located in *ATN1*, *DMPK*, and *ATXN3*—were inferred to evolve neutrally (Table 1). The *TBP* microsatellite showed evidence of selection, with support spread among different models and favoring a role for dominance (Table 1). Across these four microsatellites, our sample contained only one allele of known pathogenic size (43x) at the *TBP* locus.

All four trinucleotide microsatellites previously identified as having anomalously low variance for their allele size were found to be targets of selection, with additive models combining to receive majorities of support (Table 1). Three of these four microsatellites are in genes that fulfill a range of functions. One microsatellite is found in *GRIN2B*, which codes for a subunit of a postsynaptic receptor for the neurotransmitter glutamate and is implicated in learning, memory, and neuronal development [85]. The second microsatellite is found in the intron of an alternative transcript of the gene *DPT*, whose protein dermatopontin is part of the extracellular matrix [86] and postulated to modulate the activity of the cytokine *TGF-β* [87]. The third intragenic microsatellite is found in the gene *HRC* whose protein helps facilitate skeletal muscle contraction and has been connected to heart disease [88]. The last of the four trinucleotides is intergenic but found in an interesting location. Namely, this TTA-repeat sits at the midpoint of a ~15 kbp intergenic sequence separating tightly linked genes *ARRDC1* and *ZYMND19*, which are transcribed in opposite directions *away* from the microsatellite. In this location, the microsatellite is bracketed by the promoters of both genes, suggesting a possible co-regulatory role for the microsatellite.

All exonic dinucleotides were identified as evolving under a model of selection (Table 1). The additive, single-optimum model received overwhelming support for 13 of 18 of these loci

(Table 1). Dinucleotides in two genes—*FGFRL1* and *RBM5*—were judged to have periodic optima. The microsatellite in *SLTM* was inferred to follow a dominant, single-optimum model (Table 1). The *C3ORF27* microsatellite is only a partial exonic dinucleotide and was supported as neutral (Table 1). Only the first of its GA repeats belongs to an exon–albeit, as the open reading frame's UGA stop codon–while the remaining GA repeats belong to its 3' UTR.

The group of microsatellites known or suspected to regulate gene expression in humans displayed variable evidence of selection. Two of the eleven loci we chose from the set compiled by Rockman and Wray [39]–found in genes *COL1A2* and *SLC11A1* –showed strong signs of selection (Table 1). The *COL1A2* microsatellite is located in the first intron roughly 1 kbp downstream of exon 1; it overlaps with both a candidate cis-regulatory sequence (ENCODE; proximal-enhancer-like signature) and a strong H3K27Ac mark. The *SLC11A1* microsatellite, on the other hand, is located in the gene's promoter, approximately 330 bp upstream of the gene's transcription start site. The allele frequency distributions of these two loci are intriguing for different reasons. While the *COL1A2* microsatellite showed a strong bimodal distribution of allele frequencies (favoring the existence of periodic optima), the microsatellite in *SLC11A1* is notable for its near lack of variation (global $V_{AS}$ = 0.2). The neutral model received the greatest support for the other nine microsatellites chosen from the set in [39]. In addition, two long dinucleotides associated with signals of transcription factor binding in the form of strong DNaseI marks were both overwhelmingly resolved as neutral and associated with large $V_{AS}$ (Table 1).

One-third of intergenic dinucleotides chosen without regard to function (18 of 54) were identified as evolving under selection (Table 1). Intergenic dinucleotides possessed zero or low $V_{AS}$ in all 18 cases. Yet, each of the four selective models was favored by a subset of these microsatellites, suggesting variation in the fitness landscape among loci.

**Strength of selection spans a wide range across microsatellites.**   To characterize the strength of selection on microsatellites, a separate RF regression model was trained for each combination of motif size and model of selection (e.g., dinucleotide mutation model + additive, single-optimum). For each locus inferred to fit a model of selection, the appropriate RF model was used to estimate the strength of selection (*s*), as well as its 95% Highest Posterior Density (HPD). Like RF models trained for classification of evolutionary models, the variables of greatest importance to training regression RF models for estimating *s* were the observed heterozygosities of all eight simulated populations. For loci where $V_{AS}$ was zero or negligible, *s* was non-identifiable because any value of *s* above some threshold was sufficient to eliminate all genetic variation at the locus. In these cases, we ran 1,000 simulations to identify the minimum *s* needed to consistently obtain no variation.

The greatest point estimate of *s* = 0.013 was obtained for three loci evolving by an additive selection model: two coding dinucleotides (located in *C3ORF27* and *RBM5*) and an intronic trinucleotide (Table 2; bold-faced). Three of the four trinucleotide loci sampled for their coupling of low variance and relatively large mean allele size returned noticeably large point estimates of *s* (Table 2; triOther). Point estimates of *s* for intergenic dinucleotides identified as targets of selection ranged from 0.0013 to 0.008 (Table 2). The trinucleotide microsatellite in exon 3 of *TBP*, which codes for a poly-glutamine tract of the protein, is notable for obtaining a relatively high point estimate of *s* = 0.006 despite its reasonably high $V_{AS}$. Our results indicate substantial variation in the strength of selection targeting microsatellites.

**Estimates of key allele size comport with empirical allele frequency distributions.**   To pinpoint microsatellite alleles with the highest fitness, RF classification models for each combination of motif size and model of selection were used to classify each microsatellite by its key allele size, $\alpha$. Table 2 lists the estimated value of $\alpha$ as well as the posterior probability of that value. Like *s*, this parameter was non-identifiable for loci with zero or negligible $V_{AS}$. In these

**Table 2. Estimates of evolutionary model parameters *s* (strength of selection) and *α* (key allele size) for microsatellites receiving a plurality of support for one of the four selection models.** For loci with zero or negligible variance in allele size ($V_{AS}$), we note the minimum *s* required to maintain zero variance at a locus of the same motif size, under the identified model of selection, and using an assumed value of *α*.

| position (GRCh38/hg38) | gene | MAS | VAS | category | motif size | model | s (95% HPD) | α (posterior) |
|---|---|---|---|---|---|---|---|---|
| chr6:170561908–170562021 | *TBP* | 36.46 | 3.035 | triDisorder | trinucleotide | DPO | 0.006 (0.001, 0.0166) | 9 (0.369) |
| chr12:13774980–13775006 | *GRIN2B* | 9.06 | 0.061 | triOther | trinucleotide | APO | 0.012 (0.004, 0.0192) | 9 (0.985) |
| **chr1:168798149–168798174** | ***DPT*** | **9.98** | **0.108** | **triOther** | **trinucleotide** | **APO** | **0.013 (0.002, 0.0195)** | **10 (0.549)** |
| chr9:137597929–137597955 | intergenic | 9.08 | 0.109 | triOther | trinucleotide | ASO | 0.009 (0.002, 0.0190) | 9 (0.859) |
| chr19:49154627–49154656 | *HRC* | 11.32 | 0.298 | triOther | trinucleotide | APO | 0.005 (0.002, 0.017) | 11 (0.960) |
| chr2:218381902–218381945 | *SLC11A1* | 22.26 | 0.202 | RW | dinucleotide | ASO | 0.003 (0.0015, 0.0041) | 22 (0.921) |
| chr7:94396316–94396346 | *COL1A2* | 22.12 | 4.056 | RW | dinucleotide | DPO | 0.005 (0.001, 0.0166) | 5 (0.962) |
| chr1:181710966–181710979 | *CACNA1E* | 8 | 0 | CodingDi | dinucleotide | ASO | >0.003, non-identifiable | 8 (assumed) |
| chr1:241683527–241683540 | *WDR64* | 7 | 0 | CodingDi | dinucleotide | ASO | >0., non-identifiable | 7 (assumed) |
| chr1:42536529–42536545 | *CCDC30* | 8 | 0 | CodingDi | dinucleotide | ASO | >0.003, non-identifiable | 8 (assumed) |
| chr2:10777693–10777706 | *ATP6V1C2* | 7 | 0 | CodingDi | dinucleotide | ASO | >0., non-identifiable | 7 (assumed) |
| chr3:65387175–65387188 | *MAGI1* | 7 | 0 | CodingDi | dinucleotide | ASO | >0., non-identifiable | 7 (assumed) |
| chr4:53453081–53453094 | *FIP1L1* | 7 | 0 | CodingDi | dinucleotide | ASO | >0., non-identifiable | 7 (assumed) |
| chr5:140653008–140653021 | *IK* | 7 | 0 | CodingDi | dinucleotide | ASO | >0., non-identifiable | 7 (assumed) |
| chr5:6754900–6754913 | *PAPD7* | 7 | 0 | CodingDi | dinucleotide | ASO | >0., non-identifiable | 7 (assumed) |
| chr7:139409619–139409632 | *LUC7L2* | 7 | 0 | CodingDi | dinucleotide | ASO | >0., non-identifiable | 7 (assumed) |
| chr10:128106848–128106862 | *MKI67* | 7 | 0 | CodingDi | dinucleotide | ASO | >0., non-identifiable | 7 (assumed) |
| chr14:23059293–23059306 | *ACIN1* | 7 | 0 | CodingDi | dinucleotide | ASO | >0., non-identifiable | 7 (assumed) |
| chr16:50334767–50334780 | *BRD7* | 7 | 0 | CodingDi | dinucleotide | ASO | >0., non-identifiable | 7 (assumed) |
| chr22:35265582–35265595 | *HMGXB4* | 7 | 0 | CodingDi | dinucleotide | ASO | >0., non-identifiable | 7 (assumed) |
| chr4:1025267–1025304 | *FGFRL1* | 10 | 0 | CodingDi | dinucleotide | APO | >0.001, non-identifiable | 10 (assumed) |
| **chr3:128573476–128573493** | ***C3ORF27*** | **9.01** | **0.002** | **CodingDi** | **dinucleotide** | **ASO** | **0.013 (0.004, 0.0196)** | **9 (0.871)** |
| chr15:58890365–58890378 | *SLTM* | 6.99 | 0.007 | CodingDi | dinucleotide | DSO | >0.001, non-identifiable | 7 (assumed) |
| chr10:68749528–68749541 | *CCAR1* | 6.99 | 0.02 | CodingDi | dinucleotide | ASO | >0.001, non-identifiable | 7 (assumed) |
| **chr3:50118455–50118476** | ***RBM5*** | **10.95** | **0.093** | **CodingDi** | **dinucleotide** | **APO** | **0.013 (0.003, 0.0196)** | **11 (0.824)** |
| chr2:122694258–122694269 | intergenic | 6 | 0 | InterDi | dinucleotide | ASO | >0., non-identifiable | 6 (assumed) |
| chr4:179569040–179569052 | intergenic | 6 | 0 | InterDi | dinucleotide | ASO | >0., non-identifiable | 6 (assumed) |
| chr4:179870055–179870068 | intergenic | 7 | 0 | InterDi | dinucleotide | ASO | >0., non-identifiable | 7 (assumed) |
| chr11:23562933–23562948 | intergenic | 8 | 0 | InterDi | dinucleotide | ASO | >0.003, non-identifiable | 8 (assumed) |
| chr18:68152140–68152155 | intergenic | 8.01 | 0.005 | InterDi | dinucleotide | ASO | >0.003, non-identifiable | 8 (assumed) |
| chr2:123157113–123157132 | intergenic | 10.01 | 0.012 | InterDi | dinucleotide | APO | >0.005, non-identifiable | 10 (assumed) |
| chr13:103886021–103886035 | intergenic | 7.03 | 0.028 | InterDi | dinucleotide | DSO | 0.005 (0.001, 0.0138) | 7 (0.918) |
| chr6:23404523–23404540 | intergenic | 9.09 | 0.165 | InterDi | dinucleotide | APO | 0.007 (0.002, 0.019) | 9 (0.971) |
| chr2:220893214–220893233 | intergenic | 10.34 | 0.274 | InterDi | dinucleotide | APO | 0.006 (0.001, 0.0178) | 10 (0.935) |
| chr13:34477912–34477943 | intergenic | 13.04 | 0.425 | InterDi | dinucleotide | DPO | 0.008 (0.002, 0.0179) | 13 (0.989) |
| chr8:59646798–59646824 | intergenic | 13.2 | 0.449 | InterDi | dinucleotide | APO | 0.008 (0.002, 0.0184) | 13 (0.956) |
| chr2:22727950–22727977 | intergenic | 14.46 | 0.508 | InterDi | dinucleotide | APO | 0.005 (0.001, 0.0169) | 7 (0.871) |
| chr4:36922294–36922325 | intergenic | 15.35 | 0.93 | InterDi | dinucleotide | DPO | 0.008 (0.002, 0.0189) | 5 (0.643) |
| chr5:165163389–165163415 | intergenic | 13.82 | 1.001 | InterDi | dinucleotide | APO | 0.006 (0.005, 0.0187) | 7 (0.764) |
| chr9:11326956–11326990 | intergenic | 18.29 | 1.27 | InterDi | dinucleotide | DSO | 0.002 (0.001, 0.0026) | 19 (0.999) |
| chr5:4291710–4291743 | intergenic | 17.7 | 1.297 | InterDi | dinucleotide | DSO | 0.006 (0.001, 0.0181) | 17 (1.0) |
| chr14:86420038–86420078 | intergenic | 18.89 | 1.588 | InterDi | dinucleotide | DSO | 0.0013 (0.001, 0.0019) | 19 (0.999) |
| chr4:179230364–179230386 | intergenic | 16.12 | 24.66 | InterDi | dinucleotide | DPO | 0.006 (0.001, 0.0181) | 11 (0.969) |

cases, we assumed $\alpha$ was the closest integer to the $M_{AS}$ of the locus (Table 2). When posterior probabilities could be estimated, a single allele at each microsatellite received high support as the key allele.

The most influential summary statistics used to train models for predicting $\alpha$ included the frequency of the allele ultimately identified as $\alpha$ (or a multiple of $\alpha$, in the case of periodic-optima models) as well as the frequencies of closely adjacent alleles. Using the example of the *TBP* locus again, frequencies of 35x (hereafter, we use this shorthand to denote allele size; *e.g.* 35x indicates 35 copies of the repeat), 36x (a multiple of $\alpha$ = 9), and 37x alleles in all populations showed the greatest values of Gini decrease.

The *TBP* trinucleotide also provides a good example of: (1) the consistent agreement between estimated values of $\alpha$ and sampled data; and (2) how to interpret the estimate of $\alpha$ under one of the two periodic-optima models. Although $M_{AS}$ = 36.5 at the *TBP* locus, $\alpha$ was estimated to be 9. Intuitively, this estimate provides a good fit to the data under a periodic-optima model because marginal allelic fitness is thereby maximized at alleles of size 9, 18, 27, **36**, etc.

Estimated key allele sizes ranged from 5 to 22 (Table 2), indicating that the optimal allele size of a locus varies considerably across functional microsatellites. Furthermore, microsatellite mutation has the effect of changing the physical distance between points flanking the microsatellite. Physical alteration of the local chromatin landscape by microsatellite mutation has been shown empirically to modulate gene expression [36], validating earlier hypotheses that microsatellite length changes might act as fine-tuners of gene expression [51, 52]. In this context, estimates of fitness surfaces with multiple key alleles (see the example of the *COL1A2* microsatellite reported below) suggest that while microsatellites *in vivo* explore genotypic space, they can maintain multiple alleles of optimal length.

## Microsatellite alleles navigate complex fitness surfaces

For each microsatellite determined to evolve according to one of the four selection models, point estimates of *s* and $\alpha$ were used to reconstruct the diploid genotypic fitness surface. Fig 5A–5H shows the estimated fitness surfaces for the seven intragenic microsatellites identified as targets of natural selection. These genotype-to-fitness maps are substantially more complex than the same mapping for a diallelic SNP under an additive or dominant selective regime (Fig 5I). Our results illuminate the diversity of selective dynamics that shape microsatellite variation in humans.

## Case studies in microsatellite selection

**Dominance of a long, non-pathogenic allele creates a fitness ridge at a microsatellite that causes a triplet expansion disorder.**   *TBP* codes for TATA-box binding protein, a general transcription factor that plays a foundational role in assembly of the transcription initiation complex in eukaryotes. Pathogenic expansion of a long CAG trinucleotide repeat in exon 3 of *TBP* yields unwieldy poly-glutamine (poly-Q) tracts in the polypeptide sequence that can interrupt neuronal function, resulting in the neurologic disorder spinocerebellar ataxia 17 (SCA17) [89, 90]. In our sample, despite a very large mean allele size of ~36, very little variation was observed at this microsatellite (Table 1). Recovered parameter estimates of *s* = 0.006 and $\alpha$ = 9, under the dominant, periodic-optima model produced a fitness surface with an optimal "ridge" for all genotypes with at least one 36x allele (Figs 5A and S3), as well as ridges for other genotypes with at least one allele with size divisible by $\alpha$ = 9.

**A microsatellite in a gene involved in learning and memory has been subject to remarkably strong selection.**   *GRIN2B* codes for a subunit of a postsynaptic receptor for glutamate,

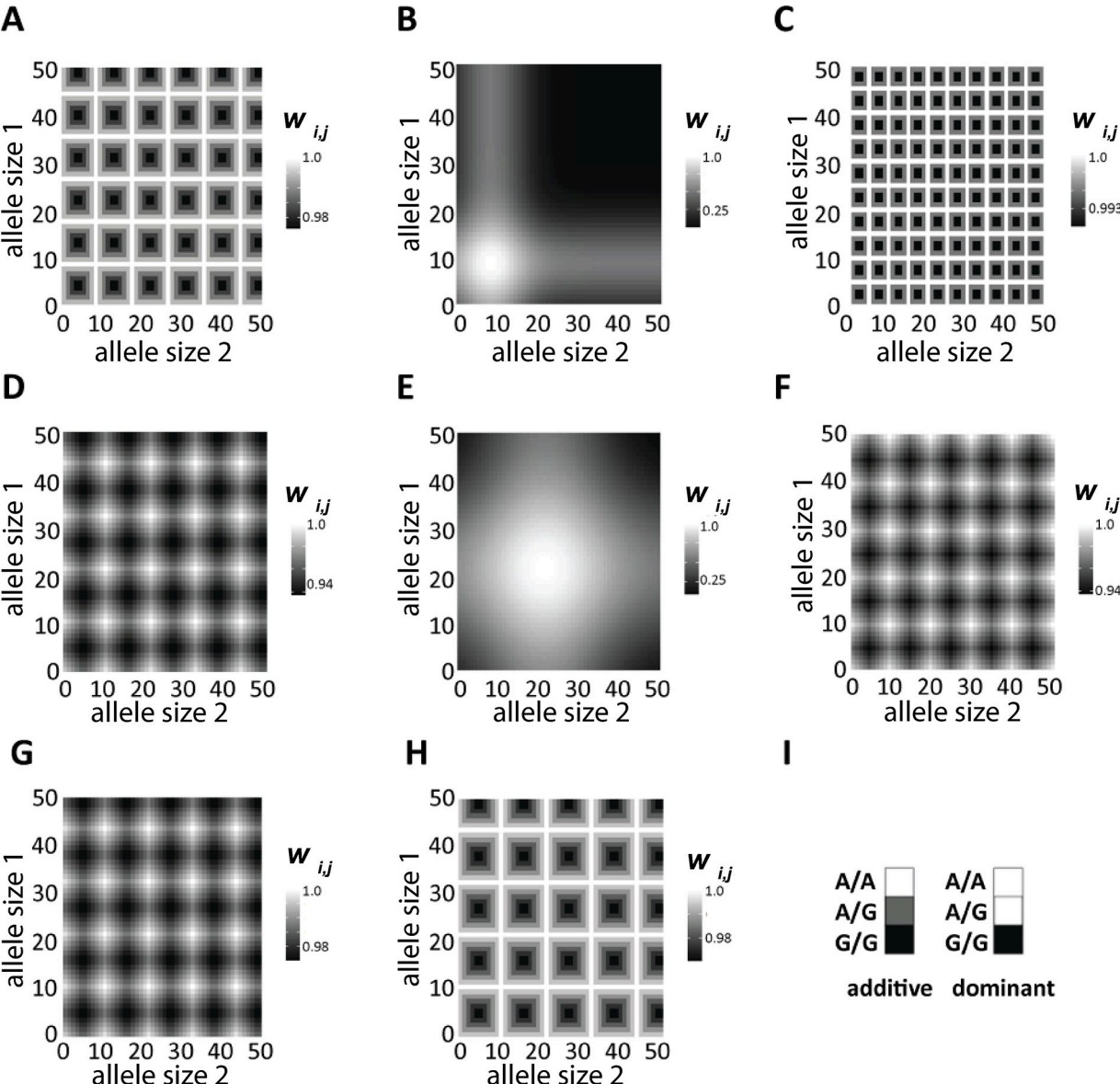

**Fig 5.** Inferred fitness surfaces for six genic microsatellites and one intergenic microsatellite in human (A-H). Note that each locus has a different scale of relative fitness ($w_{i,j}$). Examples of loci following the additive single-optimum (B, gene *GRIN2B*; E, gene *SLC11A1*), dominant periodic-optima (A, gene *TBP*; C, gene *COL1A2*; H, intergenic locus on chromosome 4), and additive periodic-optima (D, gene *RBM5*; F, gene *DPT*; G, gene *HRC*) models are shown. Panel I shows a simpler mapping of genotypes to fitness for a hypothetical A/G SNP in which the adaptive A allele is either additive or dominant. White, gray, and black colors represent high, middle, and low relative fitness, respectively.

which is implicated in learning and memory [91]. Mutations and epimutations to *GRIN2B* have been associated with risk for neurological diseases including attention-deficit/hyperactivity disorder, autism spectrum disorders, and schizophrenia [92–94]. Although the ATG-microsatellite is in a *GRIN2B* intron, almost no variation was seen in our sample; 19 in 20 sampled alleles were 9x. Very strong selection was estimated at *s* = 0.012 (Table 2). The periodic-optima model received the greatest support with an estimate of $\alpha$ = 9. The resulting fitness surface (Figs 5B and S4) shows a sloping decline in fitness moving in either direction from the key allele size of 9.

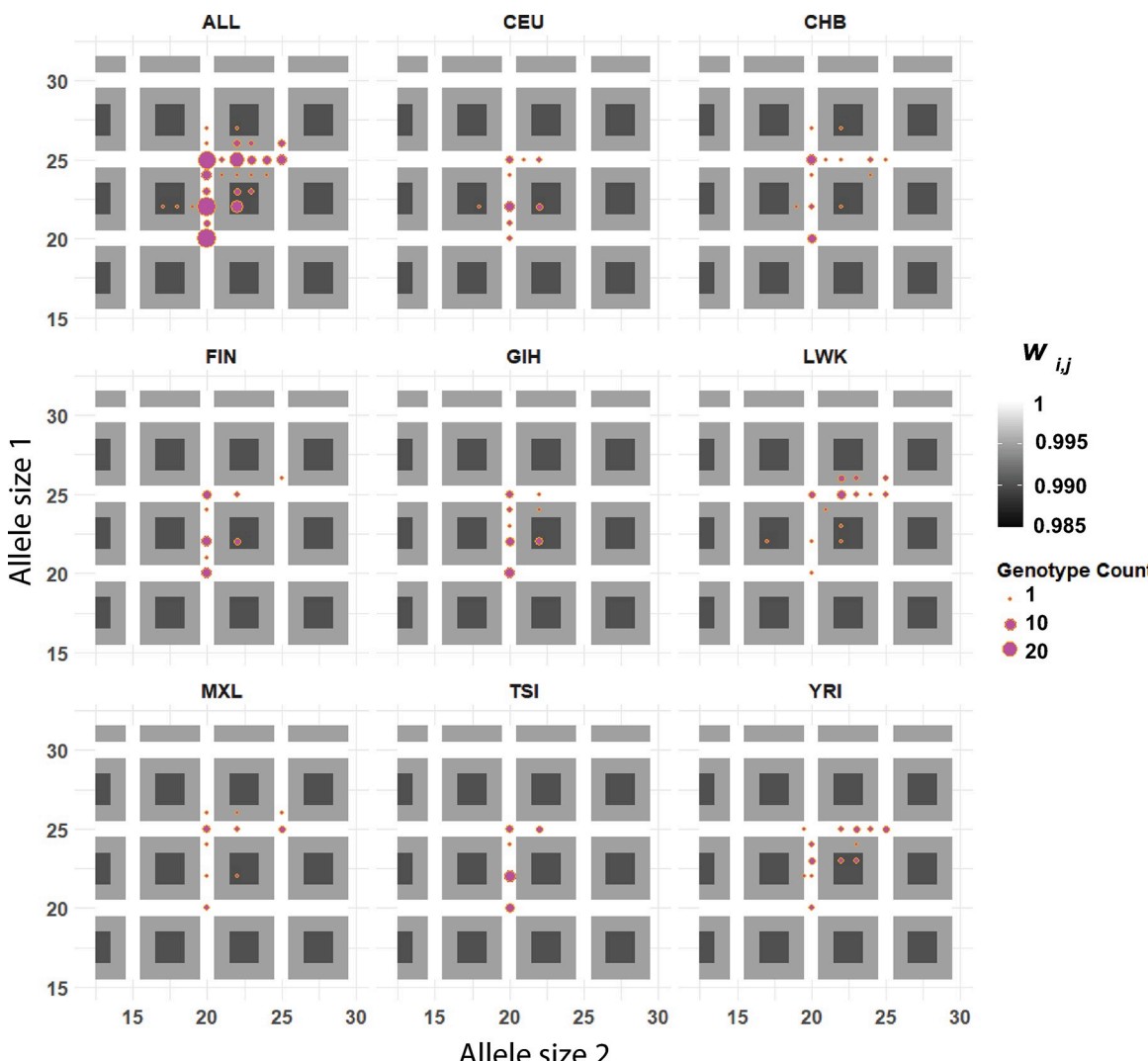

**Fig 6. Estimated dominant, periodic-optima fitness surface for a dinucleotide microsatellite in the first intron of gene *COL1A2*.**
Most genotypes contain a 20x or 25x allele in accordance with the estimated $\alpha = 5$ in a periodic optimum model. The most prominent exceptions to this pattern are in the African YRI and LWK populations.

**The fitness surface for a microsatellite in a gene that encodes collagen reflects strong dominance of two separate allele sizes.** *COL1A2* encodes one chain of type I collagen and is found in most connective tissues [95]. Mutations in this gene are associated with osteogenesis imperfecta and various subtypes of Ehlers-Danlos syndrome [96, 97]. The TG-repeat in intron 1 of *COL1A2* is coincident with a strong H3K27Ac mark and is imperfect, with 2–3 TC rather than TG dinucleotides. In the reference sequence, the longest uninterrupted TG-repeat is 14x. The *COL1A2* microsatellite received a plurality of support for the dominant periodic-optima model. Parameter estimates were $s = 0.005$ and $\alpha = 5$ (Table 2 and Fig 5C). Enrichment for genotypes with at least one allele of size 20 or 25 is evident in the fitness surfaces visualized in Fig 6, providing a good example of dominant, periodic optima with most genotypes lying along ridges of this surface that correspond to a maximum relative fitness of 1. Genotypes lacking a 20x and a 25x allele were primarily found in African populations (Fig 6).

**A microsatellite straddling the last exon and 3' UTR of a gene that encodes a nuclear RNA binding protein shows evidence of strong selection.** *RBM5* encodes a nuclear RNA binding protein that plays a role in the induction of cell cycle arrest and apoptosis [98] and is an apparent tumor-suppressor gene in lung and breast cancer [99–101]. The dinucleotide microsatellite in *RBM5* spans the last GA of the gene's 3'-most exon (in a UGA stop codon) through the beginning of its 3' UTR. The additive, periodic-optima model received a plurality of support (Table 2) at this locus and we estimated values of $s$ and $\alpha$ as 0.013 and 11, respectively; recall that $s = 0.013$ was the highest value of $s$ estimated among all loci analyzed. The 11/11 genotype is the only genotype with a relative fitness of 1.0 and therefore the sole peak of the fitness surface (S5 Fig). Alleles with sizes other than 11x at this locus are rare in our sample; only the MXL and CHB population samples included a few alleles of size 9x.

**A microsatellite known to affect expression of a gene important to human health appears subject to modest selection.** Protein SLC11A1 is a transmembrane transporter of iron and manganese. Various polymorphisms located in this gene, including the microsatellite in question, have been tied to human disease and immune response, including early-onset type 1 diabetes [102], type 2 diabetes mellitus [103], reduced immune response in tuberculosis patients [104], as well as susceptibility to tuberculosis [105]. The microsatellite in question is roughly 325bp upstream of the transcription start site of the gene and has been shown to influence the expression of *SLC11A1* [106]. Both single-optimum models received equal posterior support in model selection (Table 1). We estimated an additive, single-optimum fitness surface, in which the ubiquitous 22x allele sits atop the peak of a shallow-sloped hill (Fig 7).

**A trinucleotide microsatellite in a gene encoding an extracellular matrix protein is affected by strong additive selection.** The protein dermatopontin (DPT) is a small protein found in the extracellular matrix [86]. There is increasing evidence that DPT is involved in cell proliferation: it plays a role in wound repair [87] and, when expressed at high levels, inhibits proliferation of thyroid cancer cells [107]. The microsatellite is in the intron of a long, alternative transcript of *DPT* and bore little variation in our sample; nearly every allele sampled was a 10x repeat ($M_{AS} = 9.98$, $V_{AS} = 0.108$; Table 1). Model choice favored the additive, periodic-optima model. Strong selection was inferred with parameter estimates of $s = 0.013$ and $\alpha = 10$ (Table 2). The peak of a rapidly declining hill is found at the position of the 10/10 genotype with 10/_ heterozygous genotypes on the slopes of the surface traced to a few individuals in the African LWK and YRI populations (S6 Fig).

**Additive selection targets a trinucleotide microsatellite of a poly-glutamic acid tract in a protein implicated in the availability of calcium ion for muscle contraction.** Histidine rich calcium binding protein (HRC) is found in the lumen of the sarcoplasmic reticulum (SR) in cardiac and skeletal muscle and interacts with the protein triadin to regulate the release of calcium ions from the SR [108]. The microsatellite in exon 1 of *HRC* codes for the majority of a poly-glutamic acid (poly-E) tract of the protein HRC and mutations to this microsatellite therefore change the number of glutamic acids in the HRC's poly-E tract. The *HRC* microsatellite demonstrated greater variation ($V_{AS} = 0.298$) than the other low-variance trinucleotides of *GRIN2B* and *DPT* detailed above. However, it received strong support for the additive, periodic-optima model and parameter estimates of $s = 0.005$ and $\alpha = 11$ (Tables 1 and 2). In all populations, observed genotypes (mostly 11/11) sit huddled at the peak of a modestly sloped hill (S7 Fig).

**An intergenic microsatellite exemplifies the presence of periodic optima and the action of dominance.** An intergenic CA-repeat located on chromosome 4 provides a good example of a microsatellite in non-coding sequence that received greatest support for a model of selection. Moreover, its allele frequency distribution is clearly bimodal at allele sizes 11x and 22x (S8 Fig). The large spread between the two modes led to an estimate of $V_{AS} = 24.66$. This

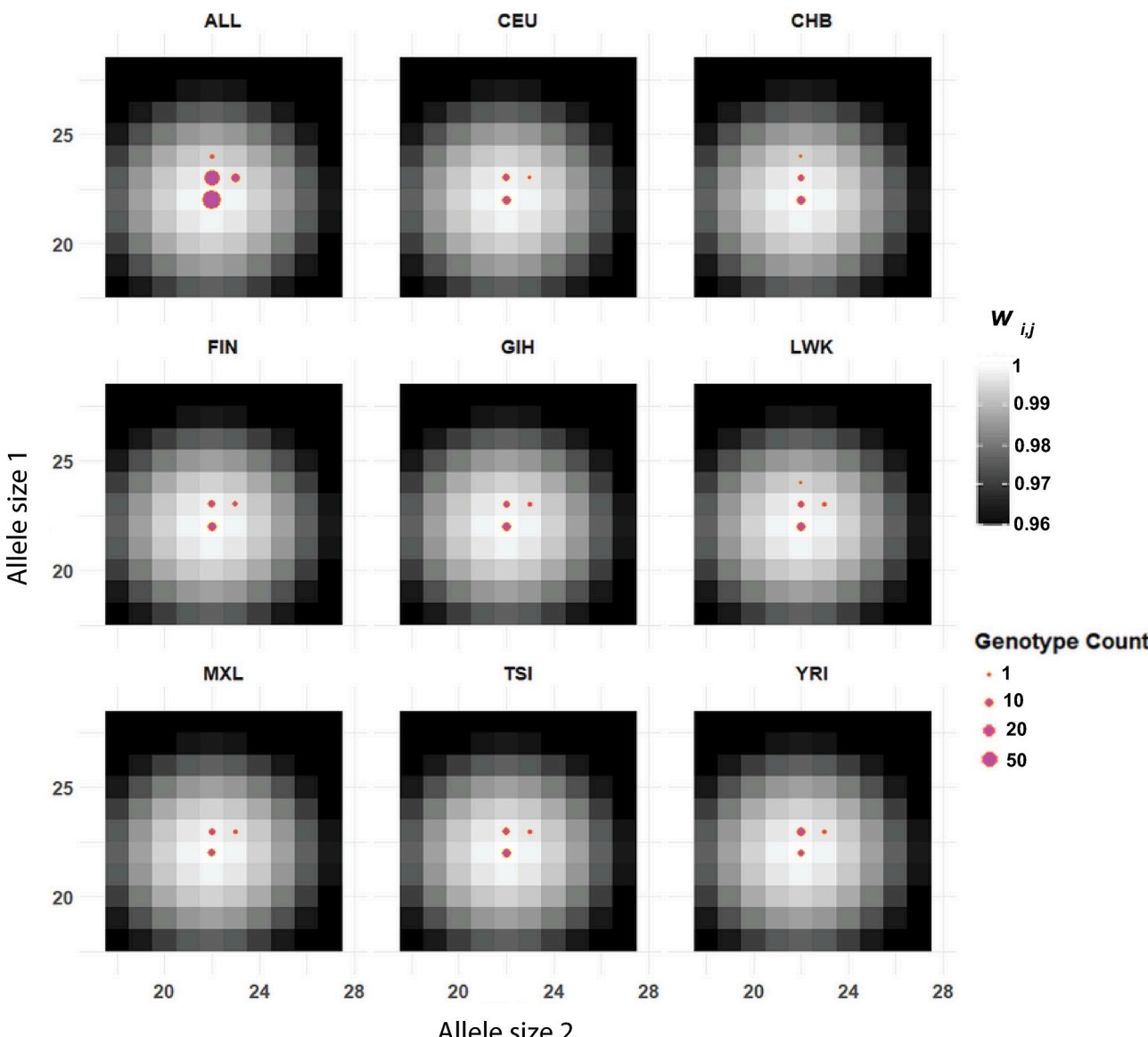

**Fig 7. Estimated additive, single-optimum fitness surface for a dinucleotide microsatellite in the basal promoter of *SLC11A1*.** Note the large allele sizes (22x, 23x, and 24x), which would suggest high mutability and predict a wider distribution of allele sizes than observed.

microsatellite received a plurality of support for the dominant, periodic-optima model, which nearly doubled support for the closest model (Table 1). This locus is a striking example in support of periodic optima models: (1) alleles cluster around genotypes that contain at least one 11x or 22x, and (2) a variety of *heterozygous* genotypes with one 11x allele that all bear a relative fitness of 1.0 due to dominance at the microsatellite.

## Discussion

We set out to address several biological and methodological questions about selection targeting microsatellites. Can we devise general models capable of capturing selection on microsatellites while accounting for their complex mutational dynamics? Given that microsatellites and other repetitive elements present a quantitatively (e.g., high mutation rate) and qualitatively (e.g.,

commonly multiallelic) different type of polymorphism than the extensively-investigated SNP–how does selection on microsatellites and other repetitive elements operate? How strong is selection on microsatellites? Is dominance a factor in heterozygous individuals? What are the fitness surfaces of microsatellites?

## Themes and lessons

Despite confirmed functional capacity of microsatellites, model-based inferences about the role of natural selection at these loci are only beginning to appear [12, 24]. The selection models reported here are applicable to different modalities of natural selection acting on microsatellites, including both positive and purifying selection. The simple equations underlying our models of selection (see Materials and Methods) provide the relative fitness of any possible genotype. These models thereby lead naturally to fitness landscapes that are easily visualized and provide an intuitive read-out of selective dynamics. Moreover, because we focus on individual loci, the common concern regarding the fitness landscape metaphor that the x and y axes are undefined and unitless is irrelevant here. The axes are well-defined as the two alleles of the genotype in units of allele size. Our portrait of microsatellite fitness surfaces reveals several themes and lessons.

Fitness landscapes of microsatellites are remarkably diverse. Each of our four selection models was favored for a subset of microsatellites. Estimated strength of selection ranged from 0.002 to 0.013 and optimum allele size ranged from 5 to 22 repeats across loci. Although our general models capture essential parameters of selection on microsatellites, specific microsatellites' genomic context or functional role may require more heavily parameterized models with the potential to produce an even greater diversity of microsatellite fitness landscapes.

Dominance is an important component of selection targeting microsatellites. Dominant models received a plurality or majority of support for several loci. Our models made the simplifying assumption that dominance is determined by the most-fit allele of a genotype. Our results—along with the potential for microsatellites to harbor many alleles—suggest that models featuring a spectrum of dominance relations among different *pairs* of alleles are worth considering in the future.

Another striking implication of our results is that microsatellite allele frequency distributions and, by implication, fitness landscapes, can be conserved across human populations. This conservation is surprising, given the diversity of environmental conditions experienced by members of our sampled populations and the potential for mutation to rapidly generate new, adaptive allele sizes. Although our results differed between African and non-African populations at a handful of loci (see below for one example), most of the microsatellites for which we found evidence of selection showed similar patterns of variation among the eight populations in our dataset. Perhaps this finding partly reflects biases in our procedure for choosing microsatellites. Still, among the loci we analyzed, selective optima appear to have been established early in human evolution and maintained for long periods of time.

The constancy of key allele size across human populations at loci inferred to be under selection contradicts the expected outcome of microsatellites acting as tuning knobs of gene expression. The rapid evolutionary response promised by microsatellites' high mutation rates is decidedly absent from most of our dataset; instead of shifting allele frequency distributions in human populations across the globe, we note *distributional* conservation. We found one notable exception. Key allele size of the promoter microsatellite in *TBP* has apparently shifted from around 34 to 36 between African and non-African populations. This finding was affirmed by ABC-RF model choice of (1) advanced models of selection in which African and non-African populations were allowed to take different values of $s$ and $\alpha$ over (2) the models of worldwide selection covered here (S2 Text).

Finally, the rate of deleterious mutation across microsatellites in the human genome is probably high. In general, low observed heterozygosity (i.e., variation) in the sampled populations was the primary determinant of whether a model of selection was chosen. Maintenance of low variation at loci with high mutation rates requires constant purging of less-fit alleles that are continuously reintroduced by mutation–a significant mutational burden. In fact, it was recently predicted that the genome-wide burden of deleterious mutations imposed by microsatellites resembles the burden generated by SNPs [24]. As previously noted [58], these details suggest analytical investigation of microsatellites may benefit from drawing an analogy between traditional models of mutation-selection balance and microsatellite evolution by natural selection.

## Microsatellites that cause disease

We considered four loci implicated in neurological disorders resulting from pathogenic expansion of a trinucleotide repeat and uncovered evidence for selection at one: the microsatellite in TATA-box binding protein (*TBP*). Expansion of the CAG/CAA repeat located in exon 3 of *TBP* causes spinocerebellar ataxia type 17 (SCA17) [89, 109], typically when allele size falls in the range of 46-55x [109]. The maximum allele size we observed was 44x. The *TBP* microsatellite is likely subject to selective pressures beyond avoidance of allele sizes prone to pathogenic expansion. As one of six general transcription factors involved in eukaryotic transcription initiation, wild-type functioning of *TBP* is of existential importance to a cell. The poly-glutamine (polyQ) tract encoded by the *TBP* microsatellite is involved in multiple protein-protein interactions including its essential binding to another general transcription factor, TFIID [110, 111]. Collectively, these details suggest that selection acting on the *TBP* microsatellite is driven by maintenance of its role in facilitating a physical connection to TFIID rather than its association with disease.

We inferred neutral evolution for the three other microsatellites causative of the trinucleotide expansion disorders SCA3 (*ATXN3*) [112], dentatorubral-pallidoluysian atrophy (DRPLA) (*ATN1*) [113], and myotonic dystrophy type 1 (DM1) (*DMPK*) [114]. Maximum allele sizes sampled at these three loci were well below the range of pathogenic allele sizes: 32x at *ATXN3* compared to allele sizes causative of SCA3 that range from 56x to 87x [112]; 26x at *ATN1* compared to alleles causative of DRPLA that range from 49x to 88x [115]; 27x at DMPK compared to causative alleles of DM1 that range from 50x to 6500x [114]. Our results suggest that wild-type alleles at these microsatellites are not subject to selective constraint.

In addition to weighing the evidence for selection on microsatellites with established connections to disease, our findings point to microsatellites that deserve to be examined in the context of disease. One of the most striking patterns we documented is the near absence of variation at an ATG microsatellite located in intron 3 of *GRIN2B*. Mutations in this gene cause *GRIN2B*-related neurodevelopmental disorder, and variants are associated with risk for schizophrenia, autism spectrum disorder, and attention-deficit/hyperactivity disorder [92–94, 116]. *GRIN2B* encodes the GluN2B subunit of the N-methyl D-aspartate (NMDA) receptor involved in learning and memory [91]. Functional studies of variants in mice indicate that GRIN2B affects neuronal differentiation, dendrite morphogenesis, synaptogenesis, circuit refinement, and synaptic plasticity [116].

Intriguingly, *GRIN2B* is predicted to be among the genes least tolerant to mutation in humans [117]. Our results suggest this status should be extended to intolerance even of microsatellite mutations in non-coding sequence. Although point mutations in introns 2 and 11 are implicated in *GRIN2B*-related disorders [91], potential functional importance of the intronic *GRIN2B* microsatellite is unclear. The microsatellite does not overlap with known CREs or

alternative splice sites. Nevertheless, selective strength at the *GRIN2B* locus is among the strongest estimated in our study, suggesting studies of pathogenesis involving *GRIN2B* should add alleles at this microsatellite to the list of variants that deserve consideration.

## Microsatellites connected to gene expression

We examined patterns of variation at 13 microsatellites for which a cis-regulatory role in modulating gene expression was established or suspected [39]. Two of these loci showed evidence for selection, with contrasting fitness landscapes.

The microsatellite located in the promoter of *SLC11A1* plays a key role in immunity. It is associated with susceptibility to tuberculosis, inflammatory bowel disease, and rheumatoid arthritis. In mice, transcription factor HIF-1 regulates allelic variation in *SLC11A1* expression by binding directly to the microsatellite during macrophage activation by infection or inflammation [118]. *SLC11A1* encodes a macrophage protein that induces expression of MHC Class II molecules, cytokines, and chemokines. The fitness surface for the *SLC11A1* microsatellite shows evidence of weak selection, with slow declines in fitness away from the key allele size of 22x.

We also uncovered evidence for selection targeting the microsatellite found in the first intron of *COL1A2*. The length of this microsatellite is connected to differences in expression levels of the linked gene [39], a collagen gene for which mutations are associated with osteogenesis imperfecta and Ehlers-Danlos syndrome [96, 97]. Our estimated fitness surface for this microsatellite is of interest for two reasons. First, we devised the periodic-optima models based on the idea that multiples of a locus-specific, key allele size would optimize physical distance between CREs, thereby leading to a specific level of gene expression. Therefore, inference of a periodic-optima model is compelling because it confirms a pattern expected under this hypothesis that a relationship exists between microsatellite allele size, CRE spacing, and gene expression. Second, the dominant model was best supported by the data for this locus. The estimated fitness surface for the *COL1A2* microsatellite overlain with observed genotype frequencies in our sample shows that most sampled genotypes bear at least one of the two observed optimal allele sizes (20x or 25x; $\alpha$ estimated as 5) and therefore line up along the ridges of the fitness surface (Fig 6).

Eleven of the thirteen loci connected to gene expression appear to evolve neutrally, suggesting that differences in expression mediated by variation at these microsatellites do not affect fitness. In addition, correlations between microsatellite length and levels of expression are identified in a variety of ways. Support for these correlations depends on the method used to detect them. For example, we might place more trust in direct functional evidence of a microsatellite's effect on gene expression than correlative *signals* such as DNase I sensitivity.

## Microsatellites with no known or suspected function

It seems reasonable to expect that randomly chosen microsatellites located far from genes would evolve without selective constraint. Surprisingly, 18 of 54 intergenic dinucleotides were identified as targets of selection by our analysis. Our results for this set of 18 intergenic loci include inference of all four selective models and a range of selection coefficients. Like the identification of highly conserved sequences in genomic regions lacking functional annotations [119], our inference of selection targeting intergenic microsatellites paves the way for discovering new functions of non-coding DNA. The prevalence of inferred selection among our essentially random sample of intergenic microsatellites suggests microsatellites may be more frequent targets of selection than previously thought.

## Caveats and prospects

Several caveats accompany our conclusions about the fitness landscapes of microsatellites. First, although our analyses incorporated major characteristics of microsatellite mutation, we ignored some features of this process. The ratio of contraction and expansion mutations is known to skew towards contraction mutations as allele size increases [11]. We did not include this detail in simulations because empirical estimates of this characteristic are lacking. In addition, we did not account for potential decreases in mutation rate for the few microsatellites we surveyed that contain imperfect repeats. Furthermore, though we incorporated direct estimates of dinucleotide mutation rate and step size probabilities from Sun et al.[11], trinucleotide mutation rate curves were less certain. For all motif sizes, we integrated over uncertainty by drawing one of many mutation rate curves at the start of each simulation (S1 Fig).

Second, the populations we surveyed were assumed to follow a demographic history previously inferred from genome-wide patterns of sequence variation. We recognize that reconstructing the history of eight populations is challenging, even in humans. True elements of human demographic history that we did not incorporate in our demographic model (S2 Fig) presumably reduced the accuracy of our conclusions. Concerns regarding this shortcoming are somewhat allayed by the fact that allele frequency distributions tended to be consistent across the sampled populations.

Third, it is possible that the signatures of selection we identified in patterns of microsatellite variation were generated by selection on linked SNPs instead of microsatellite alleles. We consider this an unlikely alternative hypothesis because microsatellite mutation occurs frequently enough to reduce linkage disequilibrium involving the microsatellite allele size to which a new, adaptive SNP allele is initially linked. Furthermore, all other alleles of this size will begin linked to the ancestral SNP allele. Therefore, the effects of linked selection on microsatellite variation should not mimic the patterns of direct selection on a microsatellite that we detect here. For example, we do not expect selection on a SNP allele to drastically reduce variation at a linked microsatellite–the very detail common to most microsatellites found to be under selection here. If there *is* variation at the microsatellite at the onset of linked selection, the adaptive SNP allele rising in frequency will become associated with many microsatellite allele sizes. If there is *no* initial variation at the microsatellite, it will remain so if under direct selection, but not because of selection for the linked, adaptive SNP allele.

Fourth, additional selective models may merit consideration. For example, our models presuppose symmetry in fitness declines moving away from the optimal allele size. For microsatellites implicated in expansion disorders, this assumption may be violated by a stark reduction in marginal fitness at the threshold between alleles in the normal range and alleles in the disease range. A mixture model of fitness landscapes for different ranges of allele sizes could be developed to better describe the dynamics of these loci.

Fifth, our results demonstrate the value of using forward-in-time (prospective) simulations, which require considerably greater computation time to perform than retrospective, coalescent simulations [75]. Among the more than 400 summary statistics used for model choice in ABC-RF, the *observed* heterozygosities of the eight sampled populations were identified as the summaries most informative about selection. Because coalescent simulations generally return allele frequencies, their output can only be used to calculate *expected* heterozygosity. Furthermore, because expected heterozygosity is a function of observed allele frequencies (all remaining summary statistics), it does not provide additional information for inference. Thus, prospective simulation not only allows us to model complex scenarios of mutation, demography, and selection; it also generates summary statistics of considerable inferential merit.

Finally, our portrait of the universe of fitness landscapes inhabited by human microsatellites is restricted to the set of loci we analyzed. In our approach, we favored careful and separate consideration of each microsatellite over the goal of describing general principles of microsatellite evolution. As advances in sequencing and bioinformatics improve the ability to genotype microsatellites from genome sequences [27, 63–66], model-based inferences of selection targeting microsatellites are beginning to be applied genome-wide [12, 24]. The insights emerging from these complementary perspectives will enable synergistic growth in our understanding of the importance of the microsatellite variant to evolution.

## Supporting information

**S1 Table. Mutation rates assumed for dinucleotide and trinucleotide repeats (per microsatellite allele per generation) for allele sizes $< 8x$.**
(DOCX)

**S2 Table.  Table A.** Dinucleotide confusion matrix. Row headings are the model simulated while column headings are the predicted model. Bold-faced numbers indicate prediction accuracy for a given model, while underlined numbers mark the most common misclassification for a given model. Although false prediction rates are high for all four selection models, the primary source of confusion is distinguishing additive from dominant selection. Detection of neutral vs. non-neutral evolution is considerably more accurate. **Table B**. Trinucleotide confusion matrix. Notation is the same as that in Table 1a. Bold-faced numbers indicate prediction accuracy for a given model, while underlined numbers mark the most common misclassification for a given model. Although false prediction rates are high for all four selection models, the primary source of confusion is distinguishing additive from dominant selection. Detection of neutral vs. non-neutral evolution is considerably more accurate.
(DOCX)

**S1 Fig. Mutation rate curves used to simulate dinucleotide and trinucleotide microsatellites.** Alleles of greater size are more mutable. Each microsatellite motif size is represented by five curves to capture uncertainty and variability in mutation rate. The most reliable curves are those for dinucleotides and tetranucleotides, where the middle curve of each set was empirically estimated by Sun et al. [11] based only on alleles with sizes within the shown bounds (vertical, dashed lines). These curves begin at allele size 8. For allele sizes $<7$, we assumed very low mutation ($\mu<10^{-6}$) for all motif sizes. The tetranucleotide mutation curves are based on estimates of Sun et al. [11] and are included for comparison.
(TIF)

**S2 Fig. Demographic model simulated.** NWACA = Non-West-African common ancestor; NACA = non-African common ancestor; MCA = MXL/CHB common ancestor, EGCA = European/GIH common ancestor; ECA = European common ancestor; WECA = West-European common ancestor. The eight extant human populations sampled are in bold face.
(PDF)

**S3 Fig. Estimated dominant, periodic-optima fitness surface for the exonic trinucleotide microsatellite in *TBP*.** The fitness surface is overlain with the observed genotype counts (magenta circles) for all eight sampled populations and the full sample (ALL). Each population shows unexpectedly low variation in allele size given the overall average allele size of ~36. The two African populations (LWK, YRI) show a distribution of distinctly lower allele sizes. Note the bright white "ridges" on the fitness surface, which correspond to all genotypes with at least

one allele whose size is divisible by 9 and that therefore have a relative fitness of 1.0.
(TIF)

**S4 Fig. Estimated additive, periodic-optima fitness surface for the intronic trinucleotide microsatellite in *GRIN2B*.** The fitness surface is focused on the only peak where observed genotypes were found in the sample. Note that heterozygous genotype counts in this figure and the following three figures are only drawn to the left of the surfaces' antidiagonals.
(TIF)

**S5 Fig. Estimated additive, periodic-optima fitness surface for the exonic dinucleotide microsatellite in *RBM5*.** The fitness surface is focused on the only peak where observed genotypes were found in the sample.
(TIF)

**S6 Fig. Estimated additive, periodic-optima fitness surface for the trinucleotide microsatellite located in an intron of an alternative transcript of the gene *DPT*.** Despite the reasonably large modal allele size of 10x, variation is limited to three individuals from the African populations of LWK and YRI out of the 200 individuals sampled worldwide. This extreme invariability led to an estimated value of selection parameter $s = 0.013$, which is the highest point estimate of $s$ found in this study (and shared by two other loci in which selection was implicated).
(TIF)

**S7 Fig. Estimated additive, periodic-optima fitness surface for the trinucleotide microsatellite underlying most of the poly-glutamic acid tract of the protein HRC.** Again, we find a relatively large modal allele size (11x) but remarkably low variation ($V_{AS} = 0.298$). Still, this is greater variation than that observed at the *GRIN2B* and *DPT* microsatellites. Therefore, the hill of the fitness surface slopes rather gently away from the peak at genotype 11/11 (see fitness gradient scale on right).
(TIF)

**S8 Fig. Estimated dominant, periodic-optima fitness surface for the intergenic CA-repeat at chr4:179230364–179230386.** Note the evidence for dominance manifested as bright, white ridges along the fitness surface. Heterozygous genotype counts are only drawn to the left of the surfaces' antidiagonals.
(TIF)

**S1 Text. Provides an example of how we combined fragment-length microsatellite genotypes and information from 1000 Genomes genotypes at the same locus to convert our genotypes to absolute allele size.**
(DOCX)

**S2 Text. Details an apparent shift in key allele size $\alpha$ between the two African populations included in the study and non-African populations at the microsatellite in *TBP*.**
(DOCX)

## Acknowledgments

This work was partly inspired by the insights of Jim Weber, who generously shared his findings of anomalous patterns of variation at certain microsatellites. We thank members of the Payseur lab for their helpful input during the project.

## Author Contributions

**Conceptualization:** Ryan J. Haasl, Bret A. Payseur.

**Funding acquisition:** Bret A. Payseur.

**Investigation:** Ryan J. Haasl.

**Methodology:** Ryan J. Haasl.

**Supervision:** Bret A. Payseur.

**Visualization:** Ryan J. Haasl.

**Writing – original draft:** Ryan J. Haasl, Bret A. Payseur.

**Writing – review & editing:** Ryan J. Haasl, Bret A. Payseur.

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
