## [Decision Letter · Decision Letter 0]

3 Oct 2024

Dear Dr Haasl,

Thank you very much for submitting your Research Article entitled 'Fitness landscapes of human microsatellites' to PLOS Genetics.

The manuscript was fully evaluated at the editorial level and by independent peer reviewers. The reviewers appreciated the attention to an important problem, but raised some substantial concerns about the current manuscript. Based on the reviews, we will not be able to accept this version of the manuscript, but we would be willing to review a much-revised version. We cannot, of course, promise publication at that time.

If you decide to revise the manuscript for further consideration at PLOS Genetics, please aim to resubmit within the next 60 days, unless it will take extra time to address the concerns of the reviewers, in which case we would appreciate an expected resubmission date by email to plosgenetics@plos.org.

To resubmit, log into your Editorial Manager account and select the option 'Revise Submission' in the 'Submissions Needing Revision' folder.

We are sorry that we cannot be more positive about your manuscript at this stage. Please do not hesitate to contact us if you have any concerns or questions.

Yours sincerely,

John K Kelly

Guest Editor

PLOS Genetics

Justin Fay

Section Editor

PLOS Genetics

Comments from AE (Kelly):

The paper “Fitness landscapes of human microsatellites” by Haasl and Payseur has been reviewed by three expert referees. All found merit and the study and recommended either major or minor revision. I concur with the referees in their evaluation. If the authors can address their suggestions, which go mainly to increasing the clarity of the presentation, a suitably revised paper would make a fine contribution to PLoS Genetics.

In addition to the comments of the referees, I would emphasize two points for the authors to consider in revisions:

I. I agree with the suggestion from reviewer 2 that the reader needs more information about the selection models before the Results start with choosing the best model for each locus. Some combination of figures 5-6 would make a good figure 1 to sit between the Introduction and Results (preceding Table 1).

II. Until reading this paper, I was not familiar with the ABC-RF method – I may not be alone in this regard. In most areas of biology, Bayesian methods are used more frequently for parameter estimation than hypothesis testing, the latter being difficult because Bayes factors are strongly dependent on priors. Is this not true with ABC-RF classification? If not, why not? Importantly, the authors get very good classification by ABC-RF when applied to simulated datasets (lines 187-200). Were the bounds on the uniform priors (lines 726-729) tuned based on the simulations or does this not matter? It may be that the authors have covered this topic in prior papers. Regardless, a paragraph (or even half a paragraph) in the Introduction of a revised paper would help the general reader to understand the method.

Reviewer's Responses to Questions

**Comments to the Authors:**

Reviewer #1: The authors present a method to test for selection on microsatellites. They then apply their method to a set of 94 microsatellite loci genotyped in 200 (human) individuals from eight populations.

Major revisions:

1. There should be some evaluation of how well the method works. On lines 713-714 the authors write that the method was applied to loci suspected of selection as well as putatively neutral intergenic loci. Of the 94 loci tested, 40 were suspected of selection and 54 were the putatively neutral intergenic loci. The method found that 27 of the 40 loci suspected of selection were under selection, and the method also found that 18 of the 54 putatively neutral loci were under selection. I appreciate that these are real data, and some of the loci suspected of selection might in fact be neutral and some of the putatively neutral loci might in fact be under selection (as the authors argue on lines 621-631). However, without an evaluation of the method where we have high confidence of the correct answer it is difficult to know what fraction of the results are true positives, false positives, true negatives, or false negatives. I suggest the authors evaluate their method by applying it on simulated data. They should apply their method both on neutrally simulated data and on data simulated under varying strengths of selection.

2. There needs to be more clarity about imperfect repeats, e.g., (AC)7(AT)(AC)3. On lines 698-700 the authors write that microsatellite genotypes were determined by fragment-length analysis and Sanger sequencing was used to corroborate these genotypes at ten randomly chosen loci. On line 639 the authors wrote that only a few of the microsatellites they surveyed contained imperfect repeats. On lines 377-378 the authors write one of their case study microsatellites was an imperfect repeat. The authors do not say (but need to) how many individuals they used Sanger sequencing to corroborate the microsatellite genotypes. If they only used a few individuals (on only 10 microsatellites) I do not see how they can confidently say they have only a few imperfect repeats, esp. since one of their case-study microsatellites was an imperfect repeat.

Minor revisions:

1. There are some problems with the Figure 2 legend that need to be fixed. Part C is mentioned twice (once as a dominant single-optimum and once as a dominant periodic-optima) while part H is never mentioned. Also, the first time part C is mentioned in the figure legend it is referred to as a dominant single-optimum, while in Figure 2, part C is clearly a periodic-optima.

2. There needs to be more discussion of the motivation of the periodic-optima models. On lines 771-773 the authors write the periodic-optima models were motivated by the frequent occurrence of bimodal allele frequency distributions among the loci we sampled and genotyped. In several figures in the paper, I see the bimodal distributions but I don’t understand why the authors think there would be many more peaks at periodic intervals. Is this model just a way to get two optima without introducing an additional model parameter?

3. In the abstract on line 29, the authors write that 43 microsatellites were found to be under selection. On line 231, they write that 45 microsatellites were found to be under selection.

Reviewer #2: In this manuscript Haasl and Payseur assess 146 “carefully chosen” microsatellite loci for their potential fitness effects. They characterize the detailed fitness landscapes of these loci assuming evolution under four different selection models compared to those that evolve neutrally. The manuscript is extremely interesting and tackles a challenging problem. While it starts off extremely well written and clear in the introduction, the main text is somewhat hard to follow. Overall while this paper should almost certainly be published, I think the authors need to reorganize and rework the text to make it more streamlined and clearer. I am suggesting a major revision so the authors can take the time to make the manuscript more readable.

Some suggestions follow:

The order of text right off the bat is a bit odd. There is some quick discussion about the distribution of the genotypes before talking about how loci were even selected. This seems out of order.

The main text should devote more time to discussing how the “carefully curated list” was carefully curated.

More up front description of the models is needed. Suggest merging Figures 6 and 7 and putting them as figure 1. These figures aren’t even really cited until the methods. The introduction should then describe the previous models published by this group (citation 58) and the main text can then provide insight into updates.

I may have missed something, but I don’t understand Fig2A given what’s written in the text. The text says a there is an optimal ridge for all genotypes with at least one 36x allele, but I see ridges for any pair where one allele is divisible by 3… what am I missing?

The discussion is similarly uneven with several short and abrupt sections followed by several reiterations of text discussed in the main. I think the discussion needs to be rewritten.

Minor comments:

Line 190 the 66% seems to give short shrift to the results which distinguish between selection and neutral much more often. Suggest some reordering.

Reviewer #3: This is a rare paper where I don't have many super detailed questions.

It is very well written and clear.

My questions are at a higher level about some of the technical details.

1. Is there a way to test the robustness of the RF approach to assumptions about the mutation model details? Another way to phrase the question is to ask how sensitive the RF method would be to uncertainty in the empirical estimates of the mutation models?

2. The authors chose a subset of loci for analysis. What prevents a genome-wide analysis of "all" loci?

3. For the loci under selection, where to they fit in terms of contributions to background selection? Are any in regions not inferred to experience much linked selection?

4. I was confused about the definition of an "x allele", such as 36x. I may have missed it but I didn't find a succinct definition in the main text. Is this simply the number of repeats?

**Have all data underlying the figures and results presented in the manuscript been provided?**

Reviewer #1: Yes

Reviewer #2: Yes

Reviewer #3: Yes

PLOS authors have the option to publish the peer review history of their article (what does this mean?). If published, this will include your full peer review and any attached files.

Reviewer #1: No

Reviewer #2: No

Reviewer #3: No

---

## [Decision Letter · Decision Letter 1]

3 Dec 2024

Dear Dr Hassl,

We are pleased to inform you that your manuscript entitled "Fitness landscapes of human microsatellites" has been editorially accepted for publication in PLOS Genetics. Congratulations!

Yours sincerely,

John K Kelly

Guest Editor

PLOS Genetics

Justin Fay

Section Editor

PLOS Genetics

Aimée Dudley

Editor-in-Chief

PLOS Genetics

Anne Goriely

Editor-in-Chief

PLOS Genetics

Comments from the reviewers (if applicable):

The referees were fully satisfied with the revision of this paper. I concur with their decisions. I have only two final suggestions:

1. The current draft has many small 2-3 sentence paragraphs that could be condensed. See, for example, lines 81-82.

2. There seems to have been some formatting errors introduced, perhaps when the manuscript was converted between file types. See line 464 for an example involving the mutation parameter.

The authors should double check the converted document before it gets sent on to the publishers.

Best wishes,

JK

Reviewer's Responses to Questions

**Comments to the Authors:**

Reviewer #1: The revisions addressed all of my concerns.

Also, I think moving the location of the Methods section improved the paper.

Reviewer #2: The authors have addressed my concerns

**Have all data underlying the figures and results presented in the manuscript been provided?**

Reviewer #1: Yes

Reviewer #2: Yes

PLOS authors have the option to publish the peer review history of their article (what does this mean?). If published, this will include your full peer review and any attached files.

Reviewer #1: No

Reviewer #2: No

**Data Deposition**

http://datadryad.org/submit?journalID=pgenetics&manu=PGENETICS-D-24-00772R1

**Press Queries**

---

## [Editor Report · Acceptance letter]

20 Dec 2024

PGENETICS-D-24-00772R1 

Fitness landscapes of human microsatellites 

Dear Dr Haasl, 

We are pleased to inform you that your manuscript entitled "Fitness landscapes of human microsatellites" has been formally accepted for publication in PLOS Genetics! Your manuscript is now with our production department and you will be notified of the publication date in due course.

With kind regards,

Anita Estes

PLOS Genetics

On behalf of:
